# The seismo-hydro-mechanical behaviour during deep geothermal reservoir stimulations: open questions tackled in a decameter-scale in-situ stimulation experiment

Amann Florian[1], Valentin Gischig[2], Keith Evans[2], Joseph Doetsch[2], Reza Jalali[2], Benoît Valley[3] Hannes Krietsch[2], Nathan Dutler[3], Linus Villiger[2], Bernard Brixel[2], Maria Klepikova[2], Anniina Kittilä[2], Claudio Madonna[2], Stefan Wiemer[2], Martin O. Saar[2], Simon Loew[2], Thomas Driesner[2], Hansruedi Maurer[2], Domenico Giardini[2],

[1] RWTH Aachen, Chair of Engineering Geology and Hydrogeology, Lochnerstrasse 4-20, 52064 Aachen, Germany

[2] ETH Zurich, Department of Earth Sciences, Sonneggstrasse 5, 8092 Zurich, Switzerland

[3] University of Neuchatel, Centre for Hydrogeology and Geothermics (CHYN), Laboratory of Geothermics and Reservoir Geomechanics, 2000 Neuchâtel, Switzerland

**Keywords:** Deep geothermal energy, EGS, induced seismicity, in-situ experiments, hydro-mechanical coupled processes in EGS,

**Abstract**

In this contribution, we present a review of scientific research results that address seismo-hydro-mechanical coupled processes relevant for the development of a sustainable heat exchanger in low permeability crystalline rock and introduce the design of the In-situ Stimulation and Circulation (ISC) experiment at the Grimsel Test Site dedicated to study such processes under controlled conditions. The review shows that research on reservoir stimulation for deep geothermal energy exploitation has been largely based on laboratory observations, large-scale projects and numerical models. Observations of full-scale reservoir stimulations have yielded important results. However, the limited access to the reservoir and limitations in the control on the experimental conditions during deep reservoir stimulations is insufficient to resolve the details of the hydro-mechanical processes that would enhance process understanding in a way that aids future stimulation design. Small scale laboratory experiments provide ~~a~~ fundamental insights~~–~~ into various processes relevant for enhanced geothermal energy, but suffer from 1) difficulties and uncertainties in upscaling the results to the field-scale and 2) relatively homogeneous material and stress conditions that lead to an over-simplistic fracture flow and/or hydraulic fracture propagation behaviour that is not representative for a heterogeneous reservoir. Thus, there is a need for intermediate-scale hydraulic stimulation experiments with high experimental control that bridge the various scales, and for which access to the target rock mass with a comprehensive monitoring system is possible. ~~Only few intermediate-scale hydro-shearing and hydro-fracturing experiments have recently been performed in a densely instrumented rock mass. No such measurements have been performed on faults in crystalline basement rocks.~~ The In-situ Stimulation and Circulation (ISC) experiment ~~currently performed~~is designed to address open research questions in a naturally fractured and faulted crystalline rock mass at the Grimsel Test Site (Switzerland~~) is designed to address open research questions, which could not be investigated in the required detail so far~~). Two hydraulic injection phases were executed to enhance the permeability of the rock mass~~ : a hydro-shearing phase and then a hydraulic fracturing phase.~~ During the injection phases the rock mass deformation across fractures and within intact rock, the pore pressure distribution and propagation and the micro-seismic response were monitored at a high spatial and temporal resolution.

1   **Introduction**
The necessity to produce carbon dioxide neutral electricity, ideally as base-load power (i.e~~.~~, 24
hours a day, year-round) and the increased aversion to nuclear power generation have motivated
global efforts to optimize methods for extracting deep geothermal energy for electricity
production. However, currently, geothermal power production is limited to distinct geological
conditions, where fluid flow rate in geothermal reservoirs carry sufficient heat (Saar, 2011)
and/or pressure for economic power generation (Randolph and Saar, 2011a; Breede et al., 2013;
Adams et al~~.,~~, 2015). It is widely agreed that the earth's crust holds substantially more
geothermal resources than are presently being exploited (e.g~~.~~, Tester et al., 2006). However,
standard water- or brine-based geothermal power generation requires persistent high reservoir
permeabilities of at least $10^{-16}$ $m^2$ (Manning and Ingebritsen, 1999) and temperatures of ideally
over about $170^oC$ (e.g., Evans, 2014; Saar, to be published in 2017), as otherwise it is not
economic. ~~When such temperatures are not present at relatively shallow depths of a couple of~~
~~kilometres, unconventional geothermal methods need to be employed. One such approach~~
~~targets~~Wells have to be drilled to at least 5 to 6 km depth into crystalline hard rock to reach
formation temperatures of approximately $170\text{-}200^oC$ in regions with standard geothermal
gradients of about $30^oC/km$, although such temperatures are often reached at shallower depth
if there is a low thermal conductivity sedimentary cover~~, thus requiring wells to be drilled to at~~
~~least 5 to 6 km depth into crystalline hard rock. The two main difficulties of implementing these~~
~~so-called~~. Presently, rotary drilling to such depths is uneconomic on a routine basis. Moreover,
at this depth permeability is often much less than $10^{-16}$ $m^2$ (e.g., Manning and Ingebritsen, 1999;
Saar and Manga, 2004, Achtziger-Zupančič et al., 2017), so that permeability has to be
artificially enhanced to permit circulation of fluids to advectively extract the heat energy
economically. Such systems are referred to as Enhanced or Engineered Geothermal Systems
(EGS), originally termed Hot Dry Rock (HDR) systems (Brown et al., 2012~~), are that 1) rotary~~
~~drilling to such depths is presently uneconomic on a routine basis and 2) permeabilities of hard~~
~~rocks at those depths are typically too low (e.g., Manning and Ingebritsen, 1999; Saar and~~
~~Manga, 2004) to enable circulation of fluids to advectively extract the heat (and pressure)~~
~~energy economically. Consequently,~~). EGS~~s~~ virtually always require hydraulic stimulation to
enhance the permeability to such a degree that economic geothermal power generation becomes
possible. However, the goal of controlling the permeability enhancement process~~sufficiently~~
~~enhancing permeability~~ has not yet been achieved in a sustained way, despite attempts since
the 1970s (Evans, 2014). Additionally, induced seismicity, which almost invariably
accompanies hydraulic stimulation because of high fluid injection pressure, can be problematic
inasmuch as it may reach felt or even damaging intensities (e.g., Giardini, 2009).
In this contribution, we focus on how a subsurface heat exchanger may be constructed between
boreholes at depth within low-permeability rock to form EGS, where a fluid, typically water or
brine, may then be circulated more easily than before. The artificially enhanced permeability
needs to be high enough to reach flow rates that are commercially relevant for power
production, depending on the subsurface working fluid. Larger permeability enhancements are
required for water or brine than for $CO_2$, as the latter can utilize lower temperatures and lower
permeabilities for economic geothermal power generation, due to its higher energy conversion
efficiency (Brown, 2000; Pruess, 2006, 2007; Randolph and Saar, 2011a, 2011b; Adams et al.,
2014, 2015; Garapati et al., 2015; Buscheck, 2016). Moreover, fluid flow should occur within
a large number of permeable fracture pathways that sweep a large surface area of the rock,
thereby providing longevity to the system and avoiding early thermal breakthrough, such as
occurred at the Rosemanowes Project (Parker. 1999) and the Hijiori Project (Tenma et al.,
2008). The construction of such systems (i.e., an artificial reservoir with sufficient permeability
for energy extraction) is one of the key research challenges for unlocking the large potential of
deep geothermal energy. The creation of a subsurface heat-exchanger between the boreholes in
the low permeability rock mass typically involves hydraulic stimulation, i.e., fluid injections,
during which the pore pressure is raised in the rock mass leading to the enhancements of
permeability of natural fractures and faults, and perhaps the creation of new fractures.
Hydraulic stimulation is inevitably accompanied by induced seismicity (e.g., Zoback and
Harjes, 1997; Evans et al., 2005a; Davis et al., 2013, Bao and Eaton, 2016), because the slip
triggered by the elevated pore pressure arising from injections may be sufficiently rapid to
generate seismic waves. In shale gas- and EGS-related stimulations, clouds of small induced
(micro-)seismic events are important monitoring tools for delineating the location, where rock
mass volume is undergoing stimulation (e.g., Wohlhard et al., 2006). Unfortunately, seismic
events induced by the stimulation injections may be large enough to be felt by local populations
and even to cause infrastructure damage (e.g., in Basel, 2006; Giardini, 2009). In the past few
years, induced seismicity has been recognized as a significant challenge to the widespread
deployment of EGS technology. From a reservoir engineering perspective, EGS faces two
competing but interrelated issues: 1) rock mass permeability must be significantly enhanced by
several orders of magnitude within a sufficiently large volume to enable sustainable heat
extraction over many years (i.e., 20 – 30 years) while 2) keeping the associated induced
seismicity below a hazardous level (Evans et al. 2014). Designing reservoir stimulation
practices that optimize permeability creation and minimize induced seismicity requires a
greatly improved understanding of the seismo-hydro-mechanical (SHM) response of the target
rock mass volume. Seismo-hydro-mechanical processes relevant for stimulation involve 1)
HM-coupled fluid flow and pressure propagation, 2) transient pressure- and permanent slip-
dependent permeability changes, 3) fracture formation and interaction with pre-existing
structures, 4) rock mass deformation around the stimulated volume due to fault slip, failure
processes and poroelastic effects, and 5) the transition from aseismic to seismic slip.
~~A~~In 2017, a decameter-scale, in-situ, stimulation and circulation (ISC) experiment ~~is currently~~
~~being~~was conducted at the Grimsel Test Site (GTS~~) in~~). Switzerland, with the objective of
improving our understanding of the aforementioned HM-coupled processes in a moderately
fractured crystalline rock mass. The ISC experiment activities aim to support the development
of EGS technology by 1) advancing the understanding of fundamental processes that occur
within the rock mass in response to relatively large-volume fluid injections at high pressures,
2) improving the ability to estimate and model induced seismic hazard and risk, 3) assessing
the potential of different injection protocols to keep seismic event magnitudes below an
acceptable threshold, 4) developing novel monitoring and imaging techniques for pressure,
temperature, stress, strain and displacement as well as geophysical methods such as ground-
penetrating radar (GPR), passive and active seismics and 5) generating a high-quality
benchmark dataset that facilitates the development and validation of numerical modelling tools.
This paper presents a literature review that highlights key research gaps concerning hydraulic
reservoir stimulation, and discusses which of the aforementioned research questions can be
addressed in our decameter underground stimulation experiment. We then provide an overview
of the ISC project that describes the geological site conditions, the different project phases and
the monitoring program.

## 2 Literature review

### 2.1 Stimulation ~~process~~by hydraulic shearing

The concept of mining heat from hot, low permeability rock at great depth was first proposed
at Los Alamos National Labs in the 1970s and was called Hot Dry Rock system (Brown et al.,
2012). They initially ~~envisaged~~envisioned creating a reservoir by applying oil and gas reservoir
hydrofracture technology to build a heat exchanger between two boreholes. Subsequent field
tests have demonstrated that hydraulic stimulation injections are effective in enhancing the
permeability of a rock mass by several orders of magnitude by producing irreversible fracture
opening, whilst also increasing the connectivity of the fracture network (Kaieda et al., 2005,
Evans et al., 2005b; Häring et al., 2008). Two different 'end-member' mechanisms commonly
appear in discussions of permeability creation processes through hydraulic injections: 1)
hydraulic fracturing as the initiation and propagation of new tensile fractures and 2) hydraulic
shearing, i.e., the reactivation of existing discontinuities in shear with associated irreversible
dilation that is often referred to as the self-propping mechanism. Hydraulic shearing is of
particular relevance for EGS as it has been shown that slip along fractures can generate a
permeability increase by up to 2-3 orders of magnitude (Jupe et al., 1992, Evans et al., 2005a;
Häring et al., 2008). If the rock mass in the reservoir is stressed to a critical level (e.g., Byerlee
1978), then a relatively small reduction of effective normal stress would be sufficient to cause
shearing along pre-existing discontinuities that are optimally-oriented for failure (Hubbert and
Rubey, 1959; Rayleigh et al., 1976; Zoback and Harjes 1997; Evans et al., 1999; Evans, 2005).
Thus, shearing and the associated permeability enhancement can occur at large distances from
the injection point, even though the causal pressure increases may be low (Evans et al., 1999;
Saar and Manga, 2003; Husen et al., 2007). In contrast, hydraulic fracture initiation and
propagation (i.e., the original concept of EGS to connect two boreholes) requires high pressures
exceeding the minimum principal stress to propagate hydro-fractures away from the wellbore.
The high pressure in the fracture may interact with natural fractures and stimulate them, leading
to leak-off (i.e., the extent of hydro-fractures is influenced by pressure losses and the existence
of pre-existing fractures). Therefore, hydraulic fracturing is often only considered relevant in
the near-field of a wellbore, where it improves the linkage between the borehole and the natural
fracture system. Rutledge et al., (2004) showed that shear activation of existing fractures and
creation of new fractures can occur concomitantly, dependent on the in-situ stress conditions,
injection pressure, initial fracture transmissivity, fracture network connectivity and fracture
orientation (e.g., McClure and Horne, 2014). Regardless of which process is dominant, the
direction of reservoir growth, and therefore, the geometry of the stimulated volume, depends to
a considerable degree on the in-situ stress gradient, stress orientation and the natural fracture
network.
Pressurized fractures may open due to a reversible compliant response to pressure (Rutqvist
1995; Rutqvist and Stephansson 2003; Evans and Meier, 2003), or due to largely irreversible
shear dilation (Lee and Cho 2002; Rahman et al., 2002). As a consequence of the coupling
between pressure, fracture compliance and permanent fracture aperture changes, the pressure
field does not propagate through the reservoir as a linear diffusive field, but rather as a pressure
front (Murphy et al., 2004). The fracture normal and shear dilation that occurs in response to
elevated fluid pressure thus has a major influence on the magnitude and profile of the
propagating pressure perturbation in the rock mass during hydraulic stimulations (Evans et al.,
1999; Hummel and Müller, 2009). As a consequence, fracture compliance and normal/shear
dilation characteristics have an impact on the size and geometry of the reservoir created during
hydraulic stimulation.
Although the aforementioned processes are conceptually well understood, the quantification
and detailed understanding required for designing stimulations and truly engineering
geothermal reservoirs are insufficient. There remains considerable uncertainty as to how the
above processes interact, and what rock mass characteristics and injection metrics control the
dominant mechanisms (Evans et al, 2005a; Jung 2013). Thermo-hydro-mechanically coupled
numerical models have become widely used for analysing relevant aspects of reservoir
stimulation in retrospective (e.g., Baujard and Bruel, 2006; Rutqvist and Oldenburg 2008;
Baisch et al., 2010; Gischig and Wiemer, 2013) or as prospective tools for predicting reservoir
behaviour or alternative stimulation strategies (e.g., McClure and Horne 2011; Zang et al.,
2013; Gischig et al., 2014; McClure 2015; Yoon et al., 2015). The fact that such numerical
models must be parameterized from sparse quantitative field-scale data is a major limitation of
all those studies. In the following we present an overview of the experimental observations of
hydro-mechanical coupling that are relevant to the parameterization of numerical models.
These stem from reservoir-scale (i.e., hectometre) stimulation operations, such as in EGS
demonstration projects or oil and gas reservoirs, intermediate-scale (i.e., decametre) in-situ
experiments, and small-scale laboratory experiments.

2.1.1  Reservoir-scale experiments
The paucity of high-quality data on the stimulation process from reservoir-scale projects is
largely because they tend to be conducted at depthsa results of the considerable depth of typical
geothermal resources (e.g., several kilometres,), which prohibits the observation of hydro-
mechanical processes from instrumentation installed within the reservoir. In the geothermal
domain, such projects constitute expensive experiments and thus are relatively few in number,
whereas, in the oil and gas domain, where hydrofracture operations are frequent and routine,
the data tend to be proprietary. Nevertheless, some notable datasets have been acquired for deep
brine injection projects (Ake et al., 2005; Block et al., 2015), deep scientific drilling projects
such as the German KTB project (Zoback and Harjes 1997; Emmermann and Lauterjung 1997;
Jost et al., 1998; Baisch and Harjes 2003), hydraulic fracturing for oil & gas production
enhancement (Warpinski 2009; Das and Zoback 2011; Dusseault et al., 2011; Pettitt et al., 2011;
Vermylen and Zoback 2011; Boroumand and Eaton 2012; van der Baan et al., 2013; Bao and
Eaton 2016;), and during the stimulation of deep geothermal boreholes (Parker, 1989; Jupe et
al., 1992; Cornet & Scotti, 1993; Tezuka & Niitsuma, 2000; Asanuma et al., 2005; Evans et al.,
2005a; Häring et al., 2008; Brown et al., 2012; Baisch et al., 2015; ). Well-documented
hydraulic stimulation datasets generally include microseismic observations as well as injection
pressures and flow rates and occasionally, tilt monitoring (Evans, 1983; Warpinski et al., 1997).
Although much information can be gained from these datasets, including imaging of
microseismic structures (Niitsuma et al. 1999; Maxwell, 2014), energy balance between
injected fluids and seismic energy release (Boroumand and Eaton 2012; Zoback et al., 2012;
Warpinski et al., 2013), and source mechanisms (Jupe et al., 1992; Deichmann and Ernst, 2009;
Warpinski and Du 2010; Horálek et al, 2010), the constraints placed on the processes are
insufficient to resolve details of the hydro-mechanical processes that underpin permeability
enhancement, flow-path linkage, channelling, or the interaction with natural fractures. Many of
these processes possibly also depend on rock type. For instance, case studies analysed by Evans
et al., (2012) support the notion that injection into sedimentary rock tends to be less seismogenic
than in crystalline rock. Moreover, it is likely that a significant part of the permeability creation
processes take place in an aseismic manner (Cornet et al., 1997; Evans et al., 1998; Guglielmi
et al., 2015b; Zoback et al., 2012).), implying that seismic monitoring may only illuminate parts
of the stimulated rock volume. In many deep hydraulic stimulation projects the rock mass is
only accessed by one or at most a few boreholes, and the structural and geological models of
the reservoir are not well defined. In general, the displacements on fractures arising from the
injection can only be directly measured where they intersect the boreholes, and deformation
occurring within the rock mass is poorly resolved.
Despite limitations in reservoir characterization and monitoring, significant insights into the
stimulation process can be gleaned from the experience from the EGS projects that have been
conducted to date. Two examples in crystalline rock are studies of stimulation-induced fault
slip and changes of flow conditions in the fracture network associated with the permeability
creation processes at the Soultz-sous-forêt (Cornet et al., 1997; Evans et al., 2005b) and the
Basel EGS projects (Häring et al., 2008). At both sites, it has been shown that permeability in
the near-wellbore region increased by 2-3 orders of magnitude. At Basel, a single initially-
impermeable fracture has been shown to take at least 41% of the flow during the 30 l/s injection
stage (Evans and Sikaneta, 2013), whereas at Soultz-sous-forêt, the stimulation of the 3.5 km
deep reservoir served to enhance the injectivity of a number of naturally-permeable fractures
(Evans et al., 2005b). These fractures tended to be optimally oriented for fault slip, as also found
elsewhere by Barton et al. (1995, 1998) and Hickman et al. (1998). At Soultz-sous-forêt, it was
possible to estimate stimulation-induced slip and normal opening of fractures that cut the
borehole by comparing pre- and post-stimulation acoustic televiewer logs (Cornet et al., 1997;
Evans et al., 2005). Shearing of fractures was also proposed as the predominant mechanism of
permeability enhancement in granite at the Fjällbacka site in Sweden, by Jupe et al. (1992),
based upon focal mechanism analysis. The above observations provide evidence of a link
between shearing and permeability changes.
An additional, important lesson from deep stimulation projects is that the stress conditions in
reservoirs may be strongly heterogeneous, and that this influences the flow field (e.g., Hickman
et al, 2000). For instance, profiles of horizontal stress orientation defined by wellbore failure
observations commonly show significant fluctuations whose amplitude varies systematically
with scale (Shamir and Zoback, 1992; Valley and Evans 2009; Blake and Davatzes, 2011), even
though that may have an average trend consistent with the tectonic stress field. Strong
deviations may occur in the vicinity of faults, indicating past fault slip and complex fault zone
architecture (Valley and Evans, 2010; Hickman et al., 2000). Similarly, the hydro-mechanical
properties of faults depend on the fault architecture, which itself depends on lithology and the
damage history accumulated over geological time (Caine et al., 2006, Faulkner and Rutter 2008;
Guglielmi et al., 2008, Faulkner et al., 2010, Jeanne et al., 2012). Within a fault zone,
permeability and compliance contrasts can vary by several orders of magnitude (Guglielmi et
al., 2008), thus complicating the predictability of hydro-mechanical responses to stimulations.
In some EGS projects, it was observed that the hydraulic communication between injection and
production boreholes may be unsatisfactory for efficient exchange of heat, either because of
high flow impedance, such as in granite rock at Ogachi, Japan, (Kaieda et al., 2005), or because
of flow channelling, as inferred from early thermal drawdown in granitic rock at Rosemanowes,
UK (Nicol and Robinson, 1990), and in granodiorite at Hijiori, Japan (Tenma et al., 2008).

2.1.2 Laboratory-scale experiments
On the laboratory-scale, considerable effort has been devoted to experiments that address the
role of effective stress changes on normal fracture opening and closure, shear dilatancy and
related permeability changes (Goodman 1974; Bandis et al., 1983; Yeo et al., 1998; Esaki et
al., 1999; Gentier et al., 2000; Olson and Barton, 2001; Samuelson et al., 2009). These
experiments have demonstrated that the relationships between fluid pressure change, fracture
opening and flow within rough natural fractures are strongly non-linear. Even though
significant progress has been made on defining permeability changes during normal opening
and shear slip on the laboratory scale, the non-linear relationships between fracture opening,
changes in effective normal stress, shearing, and the resulting permeability are yet not well
constrained (Esaki et al., 1991; Olsson et al., 2001, Vogler et al., 2015). One common approach
is to represent the fracture as two parallel plates whose separation, the hydraulic aperture, gives
the same flow rate per unit pressure gradient as would apply for the natural fracture. For parallel
plates and laminar flow, the flow rate per unit pressure gradient is proportional to the cube of
hydraulic aperture. However, for rough-walled fractures, the hydraulic aperture, $a_h$, is generally
only a fraction of the mean mechanical aperture, $a_m$ (i.e., the mean separation of two surfaces),
the fraction tending to decrease with smaller apertures, although the precise relationship is
difficult to derive from fracture geometry alone (Esaki et al., 1999; Olsson and Barton 2001;
Vogler et al., 2015). At larger mechanical apertures, limited evidence suggests that an
incremental form of the cubic law might hold such that changes in mechanical aperture give
rise to equal changes in hydraulic aperture, at least for normal loading (e.g., Schrauf and Evans,
1986; Evans et al. 1992; Chen et al., 2000). For shear-induced dilation, an additional
complication arises from channel clogging due to gouge production (e.g. ~~Lee et al., 2002).~~, Lee
et al., 2002). Particle transport through fluid flow (Candela et al., 2014) and mineralogy (Fang
et al. 2017) may additionally influence permeability changes in a complex manner. Deviations
from the cubic law also occur when flow becomes non-laminar, which tends to occur at high
flow velocities (Kohl et al., 1997), or at feed points in boreholes (e.g., Hogarth et al., 2013;
Houben, 2015).
Dilatancy associated with shearing is often expressed in terms of a dilation angle, which is a
property describing the relationship between mean mechanical aperture and slip. Dilation angle
depends on the fracture surface characteristics, the effective normal stress and the amount of
slip. Particularly important within the stimulation context is the dependence of dilation on
effective normal stress, the dilation angle tending to decrease at higher effective normal stress,
in large part because shorter wavelength asperities are sheared off (Evans et al., 1999). Thus,
shearing-induced dilation is likely to be more effective at low effective normal stress, such as
in the near field of the injection where fluid pressures are relatively high. Clearly, insights from
laboratory experiments into the relationships describing fracture dilation and permeability
changes are important for understanding field observations in EGS reservoirs (e.g., Robinson
and Brown; 1990; Elsworth et al., 2016; Fang et al., 2018), and also for parametrizing numerical
models.

### 2.1.3 Intermediate-scale experiments

In-situ experiments at the intermediate-scale (i.e., decameter-scale) serve as a vital bridge between laboratory and reservoir scales. As such, they can contribute to an improved understanding of reservoir behaviour during stimulation, and to enable up-scaling of hydro-mechanical information obtained from laboratory experiments (Jung, 1989; Martin et al., 1990; Rudquist, 1995; Schweisinger et al., 1997; Cornet et al., 2003; Murdoch et al., 2004, Cappa et al., 2006; Derode et al., 2013; Guglielmi et al., 2014; 2015). Much experience has been gained from stress testing using the hydraulic methods of hydro-fracturing (HF), hydraulic testing of pre-existing fractures (HTPF) (Haimson and Cornet, 2003), and hydro-jacking (Evans and Meier, 1995; Rutqvist and Stephansson, 1996). Hydraulic tests have been commonly used to quantify pressure-sensitive permeability changes (Louis et al., 1977), and normal stiffness in natural fractures or faults (Rutqvist et al., 1998). Evans and Wyatt (1984) estimated the closure of a fracture zone from observed surface deformations induced by drilling-related drainage of fluid pressure within the structure. Similarly, Gale (1975), Jung (1989), Martin et al. (1990), Guglielmi et al (2006), and Schweisinger et al. (2009) used borehole caliper sondes to monitor changes in fracture aperture and pressure during hydraulic jacking tests. The resulting displacements and the flow and pressure responses allowed relationships between mechanical and hydraulic aperture changes to be established and helped to constrain the fracture/fault normal compliance at larger scales.

Irreversible permeability increases arising from slip-induced dilation of natural fractures are particularly relevant for stimulation of EGS and hydrocarbon reservoirs. To study the phenomenon in-situ, Guglielmi et al. (2014) developed a novel double packer system (SIMFIP) that allows the simultaneous measurement of pressure, flow rates and 3-dimensional relative displacements occurring across a fracture isolated within the interval in response to injection. The device was successful in reactivating a fault zone in a limestone formation in Southeast France (Derode et al., 2013; Guglielmi, et al., 2015). Pressure, injection rate and 3D displacements in the SIMFIP interval were measured, together with microseismic activity, tilt and fluid pressure in the vicinity of the injection borehole. The dataset is unique, and provided quantitative insights into the relationships between (i) fault dislocation including shear and permeability changes, (ii) fault normal compliance and static friction, and (iii) slip velocities and magnitudes and their relation to aseismic and seismic slip. Recently, a similar experiment was conducted in a series of interacting complex fault zones in shale (Guglielmi et al., 2015).

Distributed pore pressure and strain sensors across the faults allowed the evolution of the
pressurized and slipped areas to be constrained, which was not previously possible. Such
experiments provide a useful methodology for advancing our understanding of the hydro-
mechanical coupled processes in complex faults.

2.2  ~~Hydraulic~~Stimulation by hydraulic fracturing ~~experiments~~
Experience gained from large scale stimulation of EGS reservoirs in crystalline rock suggests
that hydraulic shearing is the dominant mechanism for permeability creation, at least
~~remote~~several tens of meters distance from the injection point (e.g. Evans, 2014). However, the
initiation and propagation of hydraulic fractures may be an important mechanism in the near
field of the wellbore to connect the wellbore to the pre-existing fracture network in the reservoir
(Cornet and Jones, 1994). Considerable effort has been devoted to understand the initiation and
propagation of hydraulic fractures on both the laboratory and intermediate field scale.

2.2.1  Laboratory scale hydraulic fracturing experiments
Many well-controlled, small-scale laboratory experiments on hydrofracture are documented in
the literature (Jaeger 1963; Zoback et al., 1977; Warpinski et al., 1982; Bruno and Nakagawa
1991; Johnson and Cleary 1991; Song et al., 2001; Jeffrey and Bunger 2007; Bunger et al.,
2011). For such experiments, samples of various shapes (e.g., hollow cylinders and perforated
prisms) are loaded along their boundaries and the internal fluid pressure is increased until a
hydraulic fracture initiates and propagates. For some tests, transparent material like
polymethylmethacrylate (PMMA) were used to image fracture growth. Some experimental
setups include multi-material "sandwiches" to study the effect of stress contrast on hydraulic
fracture containment (Jeffrey and Bunger 2007; Warpinski et al., 1982). Others study the
interaction of propagating hydrofractures with pre-existing fractures (Zoback et al., 1977;
Meng, 2011; Hampton et al, 2015) or rock textures (Ishida 2001; Chitrala et al., 2010), the
impact of injection fluids with different viscosities (Bennour et al., 2015) or the role of stress
anisotropy (Doe and Boyce, 1989) on the geometry and orientation of generated fractures, or
the interaction between multiple fractures (Bunger et al., 2011). These laboratory studies
provide important results relevant for EGS. For instance, in the common situation where a
family of natural fractures in not normal to the minimum principal stress, injections with high
viscosity fluids (viscosity dominated regime) may help maintain tensile fracture propagation
normal to the minimum principal stress despite the presence of cross-cutting fractures (Zoback

et al., 1977), whereas low viscosity fluids (toughness dominated regime) such as water will promote leak-off into the cross-cutting natural fractures, whose permeability may be increased by shear (Rutledge et al, 2003). This leak-off will tend to limit hydrofracture propagation. Laboratory studies also give insights into the influence of shear stress shadow and transfer on hydraulic fracture growth (Bunger et al., 2011). Laboratory tests have also been essential for providing well-controlled fracture initiation and propagation datasets to benchmark hydraulic fracture simulation codes (Bunger et al., 2007).

2.2.2  Intermediate scale hydraulic fracturing experiments

Intermediate scale experiments have been performed to study initiation and propagation of hydraulic fractures. Typically, they are conducted from boreholes drilled from excavations to facilitate dense near-field instrumentation and secure good experimental control. An early example is the series of experiments that took place at the Nevada Test Site in soft, bedded volcanic tuff with high porosity and high permeability (Warpinski, 1985; Warren and Smith, 1985). The pressure, flow and fracture aperture were monitored during the experiments, and the fractures were mined back at the end of the experiments. The mine back revealed that stress contrasts were the predominant influence on hydraulic fracture containment, and that the fractures consisted of multiple fracture strands and thus differed significantly from simple shapes assumed in theoretical studies. This complexity of the fracture shape impacts the flow and pressure distribution within the propagating hydraulic fractures. Another notable series of in-situ tests on hydraulic fracture propagation within the context of coal-seam mining and block cave mine preconditioning have been performed by the hydraulic fracture group of CSIRO (Chacón et al., 2004; Jeffrey et al., 1993; 1992, 2009; Jeffrey and Settari 1995; van As et al., 2004; van As and Jeffrey 2002, 2000). The block cave mining experiments were performed in hard rock media and thus are the more relevant to EGS. Those conducted in the quartz monzonite porphyries at the Northparkes mine in Australia are probably the most detailed and densely instrumented tests executed to date, and included tiltmeter monitoring, a micro-seismic network, and pore pressure sensors as well as detailed rock mass and stress characterization (Jeffrey et al., 2009). Hydrofractures were formed with water and cross-linked gels, with coloured plastic proppants added in order to facilitate their identification once the test volume was mined back. The mapped trajectories of the hydraulic fractures exhibited complex geometries, sometimes with multiple branching and crossing of joints, veins and shear zones, with and without offset. Sub-parallel propped sections accounted for 10 to 15% of the total

fracture extent, which microseismic activity indicated was more than 40 m from the injection
point. The results demonstrate that the geometry of the fractures is much more complex than
typically obtained in small scale laboratory experiments in a homogeneous material and
uniform stress field. The complexity close to the injection point is controlled by the near-well
stress perturbation and the interaction with natural fractures and rock mass fabric.
Natural fractures have also a strong influence on the propagation of hydraulic fractures. The
propagation regime (i.e., viscosity-dominated or toughness-dominated (Detournay, 2016)) can
be controlled by the injection rate and injected fluid rheology and will have likely a strong
influence on the interaction with natural fractures and the final complexity of the hydraulic
fractures, although this has not been validated by in-situ experiment. Another relevant aspect
that has not been investigated with in-situ tests is the problem of proppant transport and
distribution within the created fractures. Indeed, in the case of hydraulic fractures, the self-
propping mechanism, which results in a permanent aperture increase, is unlikely to be effective,
and so proppant placement is necessary for insuring permanent permeability enhancement.
Finally, the nature of the microseismicity generated by hydraulic fracturing is not adequately
understood. Moment tensor analyses can offer insight into the nature of the failure in a
microseismic event (Warpinski and Du, 2010; Eyre and van der Baan, 2015). For example, they
can help resolve whether the seismic radiation is primarily generated by shear on pre-existing
fractures that are intersected by the propagating fracture, with relatively little energy generated
by the advancing mode 1 tip of the hydraulic fracture (Sileny et al, 2009; Horálek et al, 2010;
Rutledge et al., 2004).

2.3  Rock mass deformation and stress interaction
Injection of fluid into a rock mass invariably leads to deformation of the surrounding rock mass
due to poroelasticity (Biot 1941) or slip-related stress changes (McClure and Horne 2014).
Numerical studies have suggested that stress interaction between adjacent fractures can have a
significant impact on the stimulation results (e.g., Preisig et al., 2015; Gischig and Preisig
2015). In most reservoir stimulations, the microseismic clouds exhibit an oblate shape, due
primarily to the interaction between the strongly anisotropic stress field with the natural fracture
population. This tendency to form an oblate ellipsoidal shape instead of a sphere may also be
promoted by stress transfer from slipped fractures which tends to inhibit slip on neighbouring
fractures (Gischig and Preisig 2015). Schoenball et al. (2012) and Catalli et al. (2013) have
demonstrated that induced earthquakes preferably occur where stress changes generated by
preceding nearby earthquakes render the local stress field to be more favourable for slip. Similar
effects have been observed for natural earthquakes (Stein 1999). The effect becomes more
important during stimulation as time goes on, especially at the margin of the seismicity cloud.
Direct observation of deformation associated with fluid injection has been observed in several
intermediate-scale in-situ experiments. Evans and Holzhausen (1983) report several case
histories of using tiltmeter arrays to observe ground deformation above high pressure hydraulic
fracturing treatments. The results show clear evidence of self-propping of the induced fractures.
van As et al. (2004). Jeffrey et al (2009) used a tiltmeter array to monitor a hydrofracturing
treatment at the Northparkes mine in Australia. The pattern of tilting indicated the induced
fracture was sub-horizontal, which was confirmed by excavating the fracture traces. Evans and
Wyatt (1984) modelled strains and tilts occurring around a well during air drilling and found
the deformation was due to opening of a pre-existing fracture zone in response to fluid pressure
changes. Derode et al. (2013) observed tilts of $10^{-7}$-$10^{-6}$ radians some meters away from small
volume injections into a fault in limestone. In contrast, Cornet and Deroches (1990) monitored
surface tilts with a 6 instrument array during injections of up to 400 $m^3$ of slurries into granite
at 750 m depth at the Le Mayet test site in France and report no resolved signal associated with
the injections.
Rock mass deformation during stimulation injections necessarily leads to stress changes in the
rock mass. Small but non-zero residual stress changes induced by hydraulic fracturing were
measured using a stress cell by van ~~Ass~~As et al. (2004). Stress changes during injections are
recognized as playing a potentially important role in determining the pattern of fracture and slip
that develops during the injection (e.g~~.~~, Preisig et al., 2015; Catalli et al, 2013).

2.4   Seismic and aseismic slip
A significant fraction of the slip that occurs on fractures within a reservoir undergoing
stimulation may be aseismic, depending upon in-situ stress and geological conditions. That
aseismic slip has occurred is often inferred indirectly from changes in the hydraulic
characteristics of a reservoir without attendant micro-seismicity (Scotti and Cornet 1994;
Evans, 1998). Direct detection of aseismic slip is difficult as it requires relative displacements
across fractures to be resolved from borehole or near-field deformation measurements (e.g.,
Maury 1994; Cornet et al., 1997, Evans et al., 2005b). For example, Cornet et al. (1997)
compared borehole geometry from acoustic televiewer logs run before and after the 1993
stimulation at the Soultz-sous-forêt site and found that 2 cm of slip had apparently occurred

across a fracture. The cumulative seismic moment of events in the neighbourhood of the fracture was insufficient to explain the observed slip magnitude, thereby suggesting a large portion of the slip had occurred aseismically. Indeed, almost all fracture zones that were hydraulically active during the stimulation showed evidence of shear and opening-mode dislocations of millimetres to centimetres (Evans et al. 2005b).

The transition from aseismic to seismic slip was directly observed by Guglielmi et al. (2015) during fluid injection into a well-instrumented fault in limestone in a rock laboratory at 280 m depth. Some 70% of a 20-fold permeability increase occurred during the initial aseismic slip period. The transition to seismic slip coincided with reduced dilation, and the inference that slip zone area exceeded the pressurized area, suggesting the events themselves lay outside the pressurized zone. Modelling the observed slip as occurring on a circular fracture with total stress drop gave a radius of 37 m and a moment release of 65e9 Nm, far larger than the estimated seismic moment release of the order of 1e6 Nm, again indicating most slip was aseismic. Guglielmi et al. (2015) concluded that the aseismic behaviour is due to an overall rate-strengthening behaviour of the gauge filled fault and seismicity occurs due to local frictional heterogeneity and rate-softening behaviour. These results are consistent with laboratory experiments performed by Marone and Scholz (1988) on fault gauge which suggest that slip at low effective normal stresses (as anticipated in the near field of a high-pressure injection) and within thick gouge layers tends to be stable (aseismic).

Apart from these observations, aseismic slip has been mostly discussed from the perspectives of semi-analytical or numerical models. Garagash and Germanovic (2012) used a slip-weakening model to show that aseismic slip depends on the stress conditions and injection pressure. Zoback et al. (2012) used McClure's (2012) rate-and-state friction model to show that aseismic slip becomes more prominent for stress states farther from the failure limit. Using the same model, Gischig (2015) demonstrated that slip velocity depends on fault orientation in a given stress field. For non-optimally oriented faults, aseismic slip becomes more prominent and the seismicity is less pronounced for lower slip velocities and shorter rupture propagation distances. These model results suggest that aseismic slip and low slip velocities may be promoted by avoiding the stimulation of optimally oriented critically-stressed faults. Clearly, a more detailed understanding of the conditions that result in aseismic slip may be a basis for less hazardous stimulations.

2.5   Induced seismicity
Keeping induced seismicity at levels that are not damaging or disturbing to the population
continues to be a major objective for EGS (Giardini, 2009; Bachmann et al., 2011; Majer et al.,
2012; Evans et al., 2012) and other underground engineering projects (oil and gas extraction,
liquid waste disposal, gas and $CO_2$ storage). Man-made earthquakes are not a new phenomenon
(Healy et al. (1968), McGarr, 1976; Pine et al., 1987; Nicholson and Wesson, 1990, Gupta,
2003). However, the occurrence of several well-reported felt events near major population
centres has served to focus attention on the problem (Giardini, 2009; Ellsworth 2013; Davies
et al., 2013; Huw et al., 2014; Bao and Eaton, 2016). Some even led to infrastructure damage,
such as followed the Mw5.7 event in Oklahoma, USA (Keranen et al., 2013), or the suspension
of the projects (e.g., the geothermal projects at Basel (Häring et al., 2008) and St. Gallen
(Edwards et al., 2014) in Switzerland. As a consequence, a substantial research effort has been
initiated to understand the processes that underlie induced seismicity. Examples are the
numerous studies that have been performed using the high-quality seismic dataset collected
during the Basel EGS experiment. Dyer et al. (2010), Kraft and Deichmann (2014) and
Deichmann et al. (2014) analysed waveforms of the seismicity to determine reliable source
locations. Terekawa et al. (2013) used an extended catalogue of the focal mechanism solutions
of Deichmann and Ernst (2009) to estimate the stress field at Basel and to infer the pore pressure
increase required to trigger the events. Goertz-Allmann et al. (2011) determined stress drop for
the Basel seismicity and found higher stress drops at the margin of the seismic cloud than close
to the injection borehole. A similar dependency for Gutenberg-Richter b-values was found by
Bachmann et al. (2012) – lower b-values tended to occur at the margin of the seismicity cloud
and at later injection times.
There are numerous analyses of induced seismicity at other EGS sites. Pearson (1981) and
Phillips et al (1997) analysed microseismicity generated during the stimulation of the 2930 m
deep 'large Phase 1' and the 3460 m deep Phase 2 reservoirs respectively at the Fenton Hill
EGS site, New Mexico. Bachelor et al. (1983) and Baria and Green (1986) summarize
microseismicity observed during the stimulation injections into the Phase 2a and 2b reservoirs
at Rosemanowes in Cornwall, UK. Tezuka and Niitsuma (2000) examined clusters of
microseismic events generated during the stimulation of the 2200 m deep reservoir at the Hijiori
EGS site in Japan. Baisch et al. (2006, 2009, 2015) analysed data from different stages of the
stimulation of the Habanero EGS reservoir in the Cooper Basin, Australia. Calò et al. (2011)
used microseismicity generated during the stimulation of the 5 km deep EGS reservoir at
Soultz-sous-forêt to perform time-lapse P-wave tomography to infer pore pressure migration

during injection. Various authors also explored the vast induced seismicity dataset of >500'000 events recorded since the 1960s at the Geysers geothermal site, where recently also an EGS demonstration stimulation has been performed (Garcia et al., 2012; Jeanne et al., 2014). The observed seismicity was partly related to injections (Jeanne et al., 2015) and thermo-elastic stress changes (Rutqvsit and Oldenburg, 2008). Here, local variability in the stress field (Martínez-Garzón et al., 2013) and volumetric source components (Martínez-Garzón et al., 2017) were inferred from detailed analysis of injection-induced seismicity.

Another major focus of induced seismicity research has been the development of hazard assessment tools for injection related seismicity. The primary goal of these efforts is to develop a dynamic, probabilistic and data-driven traffic light system that can provide real-time hazard estimates during injections (Karvounis et al., 2014; Kiraly et al, 2016), as opposed to the traditional, static traffic light system (Bommer et al., 2006). Bachmann et al. (2011) and Mena et al. (2013) developed several statistical models and tested them in pseudo-prospective manner using the Basel seismicity dataset. More complex models including physical considerations and stochastic processes (so-called hybrid-models) were developed to include information on the reservoir behaviour and from the spatio-temporal evolution of seismicity (Goertz-Allmann and Wiemer, 2013; Gischig and Wiemer, 2013; Kiràly et al., 2016). Mignan et al. (2015) evaluated reported insurance claims arising from the Basel induced seismicity in order to infer procedures for evaluating risk based on induced seismic hazard estimates.

The Gutenberg-Richter b-value, which describes the reduction in the frequency of occurrence of events with increasing earthquake magnitude, plays a key role in induced seismic hazard analysis. Schorlemmer et al. (2005) examined the b-values of earthquakes in different stress regimes and found lower values correlated with areas of higher differential stress. Similar trends have been reported for induced seismicity (Bachmann et al., 2012), but also in tectonic earthquakes (Tormann et al., 2014; Torman et al., 2015; Spada et al., 2013) and laboratory experiments (Amitrano 2003; Goebel et al., 2012). Thus, it was hypothesized that b-values are related to local stress conditions (Scholz, 2015), or - in the context of induced earthquakes – to a combination of pressure and stress conditions. Considering standard scaling laws between magnitudes and earthquake source dimensions (i.e., slip and slipped area), it has to be expected that seismicity with high b-values may have an indirect but strong impact on permeability enhancement (Gischig et al., 2014). However, these observations have so far only been qualitatively established, as the absolute stress state within the rock volume that hosts the seismicity whose b-value is estimated has not been quantitatively determined.

Whilst the hazard associated with induced seismicity is clearly an important factor for reservoir
engineering, it should not be forgotten that the shearing of fractures and fracture zones, which
is the source of the seismicity, is a key process in the irreversible permeability enhancement
that is the objective of the stimulation injections. Furthermore, precise mapping of the 3-D
distribution of events provides an indication of the direction of fluid pressure propagation and
hence the geometry (i.e., size, shape, degree of anisotropy) of the distribution of permeability
enhancement – information that is vital for drilling subsequent well (Niitsuma et al., 1999).
Managing induced seismic hazard also requires considering the design of reservoir attributes
such as size, system impedance, and heat exchanger properties that control system longevity
(e.g., Gischig et al., 2014). Currently, few case studies consider both seismicity and the related
changes that occurred in the reservoir (e.g., Evans et al., 2005a), and relatively few studies even
report both permeability changes or well injectivity (e.g., Häring et al., 2008; Evans 2005b;
Kaieda et al., 2005; Petty et al., 2013). More work is needed to quantitatively link the spatial,
temporal or magnitude distribution of seismicity with the thermo-hydraulic-mechanical
properties of the rock mass under stimulation conditions. We believe controlled experiments
on the intermediate (in-situ test site) scale supported by laboratory-scale experiments could be
key in making progress towards this end.

2.6 Open research questionquestions in hydraulic stimulation research
Research on reservoir stimulation for deep geothermal energy exploitation has been largely
performed through laboratory observations, large-scale projects, and numerical models.
Observations of full-scale reservoir stimulations have yielded important observations.
However, the difficulty in observing the processes occurring within the reservoir under
stimulation conditions severely limits the understanding of the permeability creation processes
in a way that aids future stimulation design.
Laboratory experiments are attractive because they are controllable and readily repeatable, but
they suffer from two main limitations: 1) Upscaling results to the field-scale is affected by large
uncertainties (Gale 1993). Although there is evidence that the roughness of fresh fracture
surfaces obeys well-defined scaling over many orders of magnitude (Power and Tullis, 1991;
Schmittbuhl et al., 1995), complications arise in upscaling the aperture distribution and hence
permeability of two semi-mated rough surfaces due to the effects of damage and wear of the
asperities during shearing and gouge formation (Amitrano and Schmittbuhl, 2002; Vogler et al,
2016). 2) Laboratory tests are typically performed on single fractures in relatively homogeneous
materials and uniform stress conditions, which makes upscaling to structures with multiple
fractures such as fracture zones challenging. Similarly, hydraulic fracture propagation
behaviour is usually studied with homogenous rock samples under uniform stress, and this can
lead to an over-simplistic fracture flow and/or hydraulic fracture propagation behaviour. In an
EGS reservoir, for example, the stress may be heterogeneous on the meter to decametre-scale
(Evans et al., 1999; Valley and Evans 2009; Blake and Davatzes, 2011), and the rock mass may
contain various heterogeneities such as stiffness contrasts, fractures or faults (Ziegler et al.,
632  2015).

Because of the large uncertainties in upscaling, many numerical studies make direct (i.e., not
upscaled) use of laboratory results to parameterize HM-coupled models for EGS, because so
few field-scale relationships are available (e.g., Rutqvist, 2011; McClure, 2012; Gischig et al.,
2014). This impacts the reliability of the numerical simulation studies, because the descriptions
of the processes and the input parameter values may be inappropriate for the scale of the
simulation.
Clearly there is a need for field-scale hydraulic stimulation experiments that bridge the various
scales, and are performed with the target rock mass equipped with a comprehensive monitoring
system to capture details of the processes. Recently several intermediate-scale hydro-shearing
and hydrofracturing experiments have been performed in a densely instrumented rock mass
(i.e., Guglielmi et al., 2008, 2014 and 2015; Jeffrey et al., 2009). The hydro-shearing
experiments by Guglielmi et al. (2008) have all been in sedimentary rock types at shallow depth.
No such densely-instrumented experiments have been performed in fractured and faulted
crystalline basement rocks faults, the target rocks for most EGS, where a variety of complex
fault architectures and stress-fracture system configurations need to be investigated. The on-
going In-situ Stimulation and Circulation (ISC) experiment tries to contribute to the filling of
thisaddresses these research gap. In particular, the experiment addresses gaps, with a focus on
the following research questions: (RQ):
[RQ 1]    What is the relationship between pressure, effective stress, fracture aperture, slip,
652           permeability and storativity? (i.e., the hydro-mechanical coupled response of
653           fractures)?
[RQ 2]    How does the transient pressure field propagate in the reservoir during stimulation?
[RQ 3]    How does the rock mass deform as a result of rock mass pressurization, fracture
656           opening and/or slip?

[RQ 4] How does stress transfer inhibit or promote permeability enhancement and seismicity along neighbouring fractures?

[RQ 5] Can we quantify the transition between aseismic and seismic slip and the friction models (such as rate-and-state friction) describing slip evolution and induced seismicity?

[RQ 6] How do hydraulic fractures interact with pre-existing fractures and faults and how can the interaction be controlled?

[RQ 7] How does induced seismicity evolve along faults and fractures of different orientation?

[RQ 8] How does induced seismicity along stimulated faults compare to induced seismicity along newly created hydraulic fractures?

[RQ 9] Can we quantify the link between spatial, temporal and magnitude distribution of induced seismicity and HM coupled properties of fractures and faults?

## 3 The ISC experiment

The objective of the ISC experiment is was established to contribute in findingfind answers to the above mentioned research questions by 1) stimulating a naturally fractured crystalline rock volume at the decameter scale that is exceptionally well characterized in terms of its structural, geomechanical, and hydraulic conditions and 2) providing a dense network of sensors within the test volume so as to establish a 3D data set at high spatial resolution that will yield detailed insight into geomechanical processes associated with induced micro-earthquakes, fracture shearing, permeability creation and fluid circulation. The experiment was planned and prepared during 2015 and 2016, and executed during two series of experiments in February and May 2017. We here give a general overview of the experiment site, the main concepts, and the design of the experiment, without detailing results that are to be published in future work.

3.1 The in-situ rock laboratory

The ISC experiment is beingwas performed at the Grimsel Test Site (GTS), near the Grimsel Pass in the Swiss Alps (Figure 1Figure 1a). The GTS is owned by the National Cooperative for the Disposal of Radioactive Waste (NAGRA), and was developed as a facility to host in-situ experiments relevant to nuclear waste repository research. The facility consists of a complex of tunnels at a mean depth of 480 m that penetrate crystalline rock with well-documented structures. The rock type is considered representative for the Alpine crystalline basement that

is a main target for EGS. The test site for the ISC experiment is located in the southern part of the GTS (marked in blue in Figure 1Figure 1b) between a Tunnel that is called AU Tunnel in the west and the VE Tunnel in the east.

The rock at the GTS consists of Grimsel granodiorite and Central Aar granite. Both show an alpine foliation that strikes NE and dips steeply at ~77° towards SE. The moderately fractured rock mass is intersected by ductile and brittle shear zones, as well as brittle fractures and metabasic dykes. Within the ductile shear zones, numerous fractures that are commonly partially filled with gouge are present. Three shear zone orientations can be distinguished at the GTS (Keusen 1989). The S1 shear zones are parallel to the alpine foliation with an orientation of 142/77 (i.e., dip-direction/dip). The S2 shear zones are slightly younger than S1 and oriented with 157/75 (Keusen et al., 1989a). Shearing of the S2 structures has led to minor folding of the S1 structures (Wehrens, 2015). The youngest shear zone direction (so-called S3), have E-W strikes and southward dips (183/65), and often show evidence of dextral strike-slip movement. The target volume for the injections contains an S3 shear zone that is a fracture zone bound by two metabasic dikes on either side, and that is intersected by three ductile S1 shear zones.

## 3.2 Experimental Phases

The ISC experiment iswas divided into three phases (Figure 2). To answer all aforementioned research questions a profound understanding of the local geology, hydrogeology, stress state and rock mass properties is essential. Thus, theThe *first phase* is(2015-2016) was a pre-stimulation phase that aims to characterize the rock volume in terms of geological /and structural / stress conditions, the local stress state, hydraulic and thermal properties, and fracture connectivity., all of which is essential for the design of the experiment and the interpretation of experimental results. In addition, during the pre-stimulation phase, a monitoring system iswas established that allows capturing the seismo-hydro-mechanical response at high spatial and temporal resolution that is necessary to address the outlined research questions.. The *second phase* (February – May 2017) - the main hydroshearing and hydorfracturinghydrofracturing experiment - iswas concerned with enhancing the permeability of the rock mass with high pressure fluid injections. A *third and final phase*, (June – December 2017), the post-stimulation phase, iswas dedicated to characterize the rock mass in great detail after stimulation to quantify changes in permeability, fracture connectivity and heat exchanger properties.

3.2.1  Pre-Stimulation Phase – Rock mass characterization and Instrumentation
3.2.1.1  Boreholes, rock mass characterization and geological model
The governing aspects for designing the instrumentation of the decameter-scale ISC experiment
~~are~~were 1) a detailed understanding of the geological settings in 3-dimensions (e.g~~.~~., fracture
and fault orientation and intersections, fracture density, etc.) 2) the in-situ state of stress, 3) the
pre-stimulation hydraulic conditions, including the flow field, ~~-~~preferential fluid flow path ways
and transmissivities, 4) the borehole sections used for stimulation, 5) the type of hydraulic
injection (i.e~~.~~., hydraulic shearing or hydraulic fracturing) and ~~4~~6) anticipated quantities and
spatial distributions of strain, tilt and pressure within the rock volume during stimulation.
During the pre-stimulation phase a series of 15 cored boreholes with a length between 18 and
50 m and diameters between 86 and 146 mm ~~are~~were drilled within or about the experimental
volume (Figure 3). Three boreholes ~~are~~were dedicated to stress measurements (SBH), two ~~for~~to
the stimulation injections (INJ), four ~~for~~to geophysical characterization and monitoring (GEO),
three ~~for~~to strain and temperature measurements (FBS) and another three ~~for~~to pore pressure,
strain and temperature measurements (PRP). The boreholes ~~are~~were characterized in terms of
geologic structures~~,~~, hydraulic properties and inter-borehole connectivity ~~using various~~.
Various geological (i.e~~.~~., core logging), geophysical (i.e~~.~~., optical televiewer logs, resistivity
logs ~~using a guard resistivity sonde~~, full-wave sonic logs, ground penetrating radar (GPR)
surveys ~~with unshielded antennas~~ and active seismic measurements between the injection
boreholes) and single-hole and cross-hole hydraulic methods (i.e~~.~~., packer tests such as
pressure-pulse, constant-rate and constant head injection tests, oscillating pumping tests, and
tracer tests using various solutes, DNA-encoded nanoparticles, and heat~~).~~) were used. In
addition to borehole-based characterization methods, the experimental rock volume was
characterized using detailed tunnel maps, reflection GPR from the tunnel walls and active
seismic data acquisition between the AU and VE tunnels (Figure 1b). The trajectories of the
subsequent boreholes were chosen based on ~~the basis of~~ these preliminary geological and
hydraulic data and simplified numerical HM-coupled models (i.e~~.~~., using 3DEC, Itasca 2014)
for stimulation scenarios that ~~provide~~provided an estimate of the deformation field and pore
pressure propagation along geological structures.
The joint interpretation of ~~the above~~all geophysical ~~borehole logging,~~ geological and ~~imaging~~
~~data, tunnel mapping, core logging and hydraulic test data were~~hydrogeological observations
was used to constrain a 3D structural model of the experimental volume (Krietsch et al., 2017,
Figure 4). The 3D model illustrates the intersection of the shear zones ~~that were targeted during~~
~~the ISC experiments~~ within the experimental volume ~~(Figure 4).~~. ~~S1 shear zones (numbered~~
~~from north to south: S1.1 to S1.3) within the ISC test volume have similar orientations as the~~
~~overall foliation in the rock mass. These shear zones are characterized by an increase in foliation~~
~~intensity, and a few fractures with random distribution. The highest strains were localized in~~
~~mm-thick mylonitic bands. Due to similar appearance and orientation, no distinction between~~
~~S1 and S2 shear zones are made in the ISC volume. The experimental volume is crosscut in~~
~~east-west direction by two major (up to 1 m thick) meta-basic dykes that are separated by 2 m.~~
~~Within the ISC volume the S3 shear zones have the same orientation as the meta-basic dykes.~~
~~Thus, each of the two shear zones (here referred to as S3.1 and S3.2) is localized along the~~
~~major meta-basic dykes. Shearing of the meta-basic dykes appears to have been localized in~~
~~fine ductile shear bands resulting in biotite-rich mylonitic shear bands (i.e. 1-2 cm thick). The~~
~~dextral shearing of S3 led to a deformation of S1 faults around the meta-basic dykes (Figure 4).~~
~~Multiple persistent, partly open fractures are located between and within the meta-basic dykes~~
~~and within the host rock close to the fault. The volume between the two sheared dykes is~~
~~characterized by a high brittle fracture density (i.e. more than 20 fractures per m) compared to~~
~~the rest of the rock mass (0-3 fractures per meter; Krietsch et al., 2017). The orientations of~~
~~these fractures are shown in Figure 4. The two metabasic dykes S3.1. and S3.2, and the brittle~~
~~fracture zone between the shear zone is referred to as S3 fault zone.~~
Two major ~~(up to 1 m thick)~~ meta-basic dykes (S3.1 and S3.2) up to 1 m thick with a spacing
of 2 m crosscut the volume in east-west direction. These metabasic dykes form the boundary
of a zone with a high fracture density and partly open fractures, which together with the dykes
define the S3 shear zone. The majority of brittle fractures within and outside the S3 shear zone
are oriented parallel to the boundaries of the sheared metabasic dykes, which strike E-W in the
test volume. Very few fractures penetrate into the dykes. ~~Several quartz veins are present with~~
~~strikes of NNE to E and widths ranging from millimetres up to 30 cm. However, the lateral~~
~~extension of these quartz veins is limited to the meter range.~~

3.2.1.2 Rock mass instrumentation
In addition to a detailed characterization of the test volume for the design and interpretation of
the in-situ experiment, a dense sensor network ~~is~~was required to collect the necessary data at a
sufficient spatial resolution that ~~are~~were needed to address the previously mentioned ~~nine~~
research questions (~~i.e. research question [1 to~~ RQ1-9~~])~~.~~).~~ This includes~~:~~ pore pressure
monitoring ~~[research questions 1, 2, 6]~~, strain and tilt ~~[research questions 1, 3, 4, 5, 6]~~ and
micro-seismic monitoring ~~[research questions 4, 5, 7, 8, 9]. A major aspect governing the~~
~~detailed instrumentation.~~ Instrumentation design ~~is~~was also governed by the ~~type~~types of
hydraulic injection ~~treatment (i.e.~~treatments that were performed in the ISC experiment, i.e.,
hydraulic shearing (pressurization and reactivation of natural fractures and faults) and hydraulic
fracturing ~~or hydraulic shearing). For the ISC experiment both hydraulic fracturing (i.e.~~(i.e.,
initiation and propagation of new fractures~~) and hydraulic shearing (i.e. pressurization of~~
~~natural structures such as faults) are considered.~~).

*Pore pressure, deformations and temperature*
~~Four~~To address questions related to hydro-mechanics (RQ1), pressure propagation (RQ2) and
interaction between pre-existing and hydraulic fractures (RQ6), four pressure monitoring
boreholes (three PRP boreholes and SBH15.004; Figure 3) ~~are dedicated to the measurement~~
~~of the pressure propagation [research questions 1, 2, 6]~~were instrumented at points where they
cut relevant structures ~~within the test volume during stimulation. These boreholes are~~
~~completed with resin-grouted packer systems with fixed open intervals of few litres volume for~~
~~pressure monitoring.~~. The boreholes ~~are~~were drilled approximately normal to the strike of the
main geological features~~. The intervals are chosen to capture the pore~~ (S1 and S3 shear zones).
They were completed with cement and resin-grouted packer systems with fixed open pressure
monitoring intervals that record the pressure within fracture zones or fault zones. Pressure was
also recorded in the INJ borehole that was not ~~being injected in the test~~used for stimulation
(Figure 3) ~~by deploying~~with a straddle packer system similar to the one used for high pressure
fluid injections.~~Pore pressure was monitored using a sampling rate of 20 Hz.~~ The PRP
boreholes were also equipped with pre-stressed distributed fibre optics (FO) cables for strain
and temperature measurements. Strain recordings ~~will~~give information on the HM response to
pressurization across pre-existing fractures (~~e.g. research question 3~~RQ1), and ~~9), as well as~~
~~on~~help to detect propagation of new fractures during hydrofracturing experiment (~~e.g. research~~
~~question 6).~~RQ6). Distributed temperature measurements ~~are important for~~were used during
pre- and post-stimulation thermal tracer tests.
~~Additional~~To address research questions related to rock mass deformations (RQ3-6), three
boreholes (FBS16.001-3 in Figure 3) ~~are dedicated to the measurement of rock mass~~
~~deformation associated with hydraulic stimulation. The holes are~~were equipped with both
distributed and Fiber Bragg Grating (FBG) strain-sensing optical fibers that ~~are~~were grouted in
place. One borehole (FBS16.001) is approximately normal to the strike of the main geological
features ~~(i.e. mean strike of~~and intersects both the S3 and S1 fault zones~~, Figure 4) and thus~~
~~intersects them. One~~. Another borehole is parallel to the strike of the S3.1 fault and intersects
the S1.1 fault (FBS16.002), and one is parallel to the S1.2 faults and intersects the S3 fault zone
(FBS16.003). ~~Axial strains developed~~ The FBG sensors record axial strain across borehole
sections ~~of the boreholes~~ that span potentially active fractures or the 'intact' rock mass ~~between~~
~~them are measured with FBG sensors that have an operating range of -1000 to 2000 µε and a~~
~~resolution of 0.1 µε. The objective to measure strain parallel to fault zones is to capture the~~
~~strain field that is associated with fault shearing during stimulation. Strain sensors across~~
~~structures allow quantifying the fracture dislocation.~~. Distributed strain-sensing optical fibers
allow a dense spatial coverage and thus ~~increase the likelihood~~are more likely to observe the
propagation ~~direction~~ and opening of a hydraulic fracture. ~~A parallel distribution of untensioned~~
~~Bragg Grating sensors is used to correct the strains for temperature. All FBG sensors are~~
~~monitored with a 16 channel si255 Hyperion FBG interrogator (Micronoptics), which is able to~~
~~record strain or relative temperature from more than 10 sensors per channel with sampling rates~~
~~of up to 1000 Hz. By averaging up to 1000 samples the strain resolution can be improved to~~
~~<0.1 µε. All three FO boreholes are also equipped with a distributed pre-stressed fiber optics~~
~~cable for strain and temperature monitoring that are recorded with a DiTest device from~~
~~ominsense~~.
The borehole strain monitoring system ~~is~~was complemented with an array of 3 biaxial tiltmeters
installed on the margins of the test volume along the VE tunnel near the S3 fault zone (Figure
3). ~~They are~~The tilt sensors were mounted in shallow holes drilled into the tunnel floor. ~~The tilt~~
~~sensors are of type 711-2 from Applied Geomechnics,~~ and ~~have a resolution of 0.1~~
~~µradians.~~record horizontal tilt. Together, the tilt measurements and the longitudinal strain in
the FO boreholes ~~will~~were capable to describe the deformation field around the stimulated rock
volume and ~~allow~~allowed constraining the characteristics of the stimulated fault ~~zone~~zones
(i.e~~.~~., dimension, dislocation direction and magnitude, etc~~.), which helps answering research~~
~~questions 3, 4, 5, and 9.~~.).

*Micro seismicity*

Microseismicity is monitoredQuestions related to induced seismicity (RQ5, 7, 8) were tackled using a microseismic monitoring system, which consists of a sensor network with 14 piezo sensors (Type GMuG Ma-Bls-7-70)piezosensors affixed to the tunnel walls, and 8 sensors (type GMUG Ma-Bls-7-70)that were pressed pneumatically against the borehole wall in the geophysical monitoring boreholes (GEO16.001 – 4, FigureFigures 3 and 6). The distribution of sensors within and about the experimental volume ensures optimal azimuthal and vertical coverage around the stimulation points.5). The uncalibrated piezo sensors arepiezosensors were complemented with calibrated accelerometers (Type Wilcoxon 736Tas done by Kwiatek et al., 2011) at five locations on the tunnel surface to enable the calculation of absolute magnitudes. The piezo sensors are sensitive to strain signals in the range of 1-100 kHz, while the accelerometers are sensitive from 50 Hz to 40 kHz. Signals from all sensors were recorded continuously on a 32-channel acquisition system (provided by Gesellschaft für Materialprüfung und Geophysik, GMuG) at a sampling rate of 1 MHz. An A real-time event detection, first arrival determination and location algorithm with automatic picking of first arrivals allows real time computation ofgave provisional event hypocentres. More detailed processing of the complete data is performed after the experiment (Gischig et al., 2017). Recorded induced seismicity is the basis to answer research question 5, 7 and 8.

The sensor network iswas also used to recorded periodic active seismic experiments. Highly reproducible sources (i.e., piezoelectric pulse sources in boreholes and hammers installed at the tunnel walls with pre-defined constant fall height, Figure 6) are5) were triggered roughly every 10 minutes during the stimulation experiments with the goal of recording systematic changes in the waveform characteristics that allow inferring changes of seismic velocity, attenuation and scattering properties. Such measurements can give additional constraints on 3D pressure propagation and deformation characteristics (research question 1 – 4 andRQ1-4, 9).

3.3  Stimulation Phase

As both hydroshearing and hydrofracturing are part of above research questions, theThe stimulation experiments consist ofwere performed in two parts: 1)experiment sequences: 1) In February 2017, six hydraulic shearing experiments were performed including high-pressure water injection into existing faults or fracture zones within the test volume so that theso as to reduce effective normal stress on the structures is reduced and hydraulictrigger shearing is triggered, and. 2) In May 2017, six hydraulic fracturing experiments were conducted with high

pressure injection into fracture-free borehole intervals so as to initiate and propagate hydraulic
fractures.
Two 146- mm diameter, downwardly-inclined boreholes (INJ 1 and INJ 2 in Figure 3) ~~are~~were
dedicated for the ~~hydraulic shearing and hydraulic fracturing stimulation~~ injections from
packer-isolated intervals. For the stimulation operations, water or gel ~~is~~was injected into a 1-2
m interval in one borehole, and the second borehole ~~is~~was used to additionally monitor the fluid
pressure response~~, together with other dedicated pressure monitoring boreholes.~~. The maximum
injected volume for the stimulation at each interval ~~is~~was limited to about 1000 liters~~, in order~~.
This value was determined as part of a pre-experiment hazard and risk study (Gischig et al.,
2016) and was found to ~~minimize the~~be acceptable regarding the estimated likelihood of
inducing seismic events that could be felt in the tunnels, as well as ~~avoid~~the disturbance to on-
going experiments elsewhere in the GTS. We used standardized injection protocols for HS and
HF (i.e., we did not test different injection strategies) so that the variability in the rock mass
response arises from differences of local hydromechanical conditions as well as geological
settings, and not from different injection strategies.

3.3.1  Hydroshearing ~~experiment~~experiments
The stimulation injections ~~target~~targeted natural fracture zones in the rock volume ~~whose~~
~~transmissivities ranges from 1e-8~~7 ~~to 1e-11~~10 m²/s. Each interval stimulation ~~consists~~consisted
of ~~three~~four cycles (Figure ~~7~~6). The objective of the first cycle ~~is~~was to measure initial
transmissivity and jacking pressure, and break down the interval. Initially (Cycle 1.~~1~~), pressure
~~needs to be~~was increased in small steps until breakdown ~~occurs~~occurred, as evidenced by a
disproportionate increase in flow rate. This first ~~sub-~~cycle ~~allows to quantify~~allowed
quantifying the initial injectivity. After venting, the test ~~needs to be~~was repeated with refined
pressure steps (Cycle ~~1.~~2) in a narrow range to identify the jacking pressure. After Cycle ~~1~~2
the interval ~~is~~was shut-in to capture the pressure decline curve before the interval ~~is~~was vented.
The purpose of the ~~second~~third cycle ~~is~~was to increase the extent of the stimulation away from
the injection interval. For this purpose, a step-rate injection test with four or more steps ~~is~~was
utilized ~~with a maximum rate of 37 l/min.~~. The interval ~~is~~was then shut-in and the pressure
decline ~~is~~was monitored for 40 minutes before initiating venting for 30 minutes. The purpose
of the ~~third~~fourth cycle ~~is~~was to determine post-stimulation interval transmissivity and jacking
pressure for comparison with pre-stimulation values. Thus, a step-pressure test ~~is~~was conducted
initially taking small pressure steps to define the low- pressure Darcy trend and the deviation
from it ~~that occurs at~~defining the jacking pressure. Following this cycle, the interval ~~is~~was shut-
in for 10 minutes before venting. An important aspect for the quantification of irreversible
changes in the reservoir ~~is~~was to run acoustic televiewer logs across each interval before and
after the stimulation to attempt to resolve any dislocation that may occur across the fractures in
the interval.

3.3.2  Hydraulic fracturing experiment
The protocol for hydraulic fracturing tests in borehole intervals without natural fractures ~~are~~is
shown in Figure ~~8. Again, each~~7. Each interval stimulation ~~consists~~consisted of three cycles.
First, the packed interval ~~is~~was tested with a pulse for integrity. ~~The measured transmissivity~~
~~in intact rock ranges from 1e-13 to 1e-14 m²/s.~~ The objective of the first cycle ~~is~~was to break
down the formation (i.e~~.~~, to initiate a hydraulic fracture) using small flow rates (i.e~~.~~, around 5
l/min injections for 60 s). The second cycle ~~aims~~aimed to propagate the hydraulic fracture away
from the ~~well bore~~wellbore and connect to the pre-existing fracture network using progressively
increasing flow rates (up to 100 l/min). ~~A shut-in and venting period follows~~followed. ~~Finally,~~
~~t~~The purpose of the third cycle ~~is~~was to quantify the final injectivity and jacking pressure using
a pressure step ~~injections~~injection similar to the pressure step injection considered for cycle ~~3~~4
in the fault slip experiments. Both pure water and a gel (i.e~~.~~, a Xanthan-water-salt-mixture with
0.025 weight percent of Xanthan and 0.1 weight percent of salt with a viscosity between 35 and
40 cPs) ~~are~~were used for fracture propagation. ~~If gel is~~was used, cycle 2 is extended with a~~
~~flushing cycle (with water) after fracture propagation.~~ The two injection fluids ~~allow~~allowed
investigating two different propagation regimes (i.e., toughness-dominated and viscous-
dominated). ~~A specific amount of salt was added to each injection fluid as a tracer, to investigate~~
~~flow paths and dilution effects.~~ Further, a cyclic injection sequence ~~is~~was included ~~into~~in the
fracture propagation cycle to test ~~it~~as an alternative injection protocol as proposed by Zang et
al. (2013). ~~They propose~~proposed that using cyclic injection the same efficiency in fracture~~
~~propagation can be reached, while the associated micro-seismic event release is limited and~~
~~fracture branching is enhanced.~~

3.4  Post-Stimulation Phase
~~The purpose of this~~In the last experiment phase ~~is to determine,~~ the changes to the hydrology
and rock mass properties that occurred ~~as a result~~because of each of the two stimulations phases
(i.e~~.~~, the hydraulic shearing and hydraulic fracturing phases~~)~~.) were investigated. Accordingly,

after each phase, a characterization program was performed. The hydraulic properties of the rock mass were determined using single-hole and cross-hole hydraulic methods similar to pre-stimulation the characterization phase. ~~Selected stimulation intervals were isolated with packers and then subjected to a variety of tests including pressure-pulse, constant-rate and constant head injection tests, oscillating pumping tests, and tracer tests using solute dyes, DNA-tagged nanoparticles and heat.~~ In addition, single hole, cross-hole, and cross-tunnel active seismic and GPR measurements were conducted. ~~Repeat geophysical borehole logs were run in both injection boreholes, including focused resistivity, and full-wave sonic.~~

## 4 Summary and Conclusion

The review of scientific research results showed that carefully analyzed data from large-scale experiments (i.e., EGS projects) and laboratory scale experiments provide a fundamental understanding of processes that underpin permeability creation and induced seismicity in EGS. The results from large-scale experiments suffer from accessibility and resolution, which does not permit to resolve the details of seismo-hydro-mechanical coupled processes associated with the stimulation process. Laboratory scale experiment provide a fundamentally improved understanding of these processes but suffer from scalability and test conditions that may lead to over-simplistic fracture flow and/or hydraulic fracture propagation behavior that is not representative for a heterogeneous reservoir. Intermediate-scale experiments can serve to bridge the gap between the laboratory and the large scale and may enable upscaling of results gained from small scale experiments. However, only few intermediate-scale hydro-shearing and hydro-fracturing experiments have recently been performed in a densely instrumented rock mass and no such measurements have been performed on faults in crystalline basement rocks.

We have provided here an overview of the intermediate scale hydroshearing and hydrofracturing experiment (i.e., ISC experiment) ~~is being~~that was executed in 2017 in the naturally fractured and faulted crystalline rock mass at the Grimsel Test Site (Switzerland). It ~~is~~was designed to fill some of the key research gaps and thus contribute to a better understanding of seismo-hydro-mechanical processes associated with the creation of Enhanced Geothermal Systems. As this contribution is meant to only provide a literature review and an overview of our ISC experiment at the Grimsel Test Site, several other publications will provide more detailed descriptions and analyses of this intermediate-scale hydroshearing and hydrofracturing experiment.

## 5  Acknowledgment

The ISC is a project of the Deep Underground Laboratory at ETH Zurich, established by the Swiss Competence Center for Energy Research - Supply of Electricity (SCCER-SoE) with the support of the Swiss Commission for Technology and Innovation (CTI). Funding for the ISC project was provided by the ETH Foundation with grants from Shell and EWZ and by the Swiss Federal Office of Energy through a P&D grant. Hannes Krietsch is supported by SNF grant 200021_169178. The Grimsel Test Site is operated by Nagra, the National Cooperative for the Disposal of Radioactive Waste. We are indebted to Nagra for hosting the ISC experiment in their GTS facility and to the Nagra technical staff for onsite support. We also thank two anonymous reviewers for their valuable input.

## ~~5~~6  References

Achtziger-Zupančič P., Loew S. and Mariéthoz G. (2017). A new global database to improve predictions of permeability distribution in crystalline rocks at site scale. Journal of Geophysical Research. Solid Earth, 122 (5): 3513-3539, Washington, DC: American Geophysical Union.

Adams, B.M., T.H. Kuehn, J.M. Bielicki, J.B. Randolph, and M.O. Saar (2014). On the importance of the thermosiphon effect in CPG (CO2 Plume Geothermal) power systems, Energy, DOI: 10.1016/j.energy.2014.03.032, 69:409-418.

Adams, B.M., T.H. Kuehn, J.M. Bielicki, J.B. Randolph, M.O. Saar (2015). A Comparison of Electric Power      Output of $CO_2$ Plume Geothermal (CPG) and Brine Geothermal Systems

for Varying Reservoir      Conditions,      Applied      Energy,      DOI:
10.1016/j.apenergy.2014.11.043, 140:365–377.
Ake J, Mahrer K, O'Connell D, Block L. (2005). Deep-Injection and Closely Monitored
Induced Seismicity at Paradox Valley, Colorado. Bulletin of the Seismological Society of
America, 95(2), 664–683. doi:10.1785/0120040072
Amitrano, D., (2012). Variability in the power-law distributions of rupture events, Eur. Phys.
J. Spec. Top., 205, 199–215.
Amitrano, D., and J. Schmittbuhl (2002), Fracture roughness and gouge distribution of a granite
shear band, *J. Geophys. Res.*, *107*(B12), 2375 doi:10.1029/2002JB001761.
Asanuma H, Soma N, Kaieda H, Kumano Y, Izumi T, Tezuka K, et al. (2005). Microseismic
monitoring of hydraulic stimulation at the Australian HDR project in Cooper Basin. In
Proceedings World Geothermal Congress (pp. 24–29).
Bachmann, C., S. Wiemer, B. P. Goertz-Allmann, J. Woessner (2012). Influence of pore
pressure on the size distribution of induced earthquakes, Geophysical Research Letters, 38,
L09308.
Bachmann, C., S. Wiemer, J. Woessner, S. Hainzl (2011). Statistical analysis of the induced
Basel 2006 earthquake sequence: Introducing a probability-based monitoring approach for
Enhanced Geothermal Systems, Geophys. J. Int.
Baisch, S., Vörös, R., Rothert, E., Stang, H., Jung, R. Schellschmidt, R., (2010). A numerical
model for fluid injection induced seismicity at Soutz-sous-Forêt, Int. J. Rock Mech. Min. Sci.,
47, 405–413.Baisch, S., Harjes, H.P. (2003). A model for fluid-injection-induced seismicity at
the KTB, Germany. Geophysical Journal International 152, 160–170.
Baisch, S., R. Vörös, R. Weidler, D. Wyborn (2009). Investigation of fault mechanisms during
geothermal reservoir stimulation experiments in the Cooper Basin (Australia), Bull. Seismol.
Soc. Am. 99, no. 1, 148–158.
Baisch, S., R. Weidler, R. Vörös, D. Wyborn, L. DeGraaf (2006). Induced seismicity during
the stimulation of a geothermal HFR reservoir in the Cooper Basin (Australia), Bull. Seismol.
Soc. Am. 96, no. 6, 2242–2256.
Baisch, S., Rothert, E., Stang, H., Vörös, R., Koch, Ch., McMahon, A. (2015). Continued
Geothermal Reservoir Stimulation Experiments in the Cooper Basin (Australia). Bulletin of the
Seismological Society of America, Vol. 105, No. 1, pp. 198–209
Bandis S., A.C. Lumsden, N. R. Barton (1983). Fundamentals of rock joint deformation.
International Journal of Rock Mechanics Mining Sciences & Geomech Abstr., 20, 6: 249–268.
Bao X., Eaton D. W. (2016). Fault activation by hydraulic fracturing in western Canada.
Science 10.1126/science.aag2583
Baria, R., and A. S. P. Green (1986), Seismicity induced during a viscous stimulation at the
Camborne School of Mines Hot Dry Rock Geothermal Energy project in Cornwall, England,
paper presented at 8th Int. Acoustic Emission Symp., Japanese Soc. of NDI, Tokyo, Japan,
October.
Barton C.A., M. D. Zoback, D. Moos (1995). Fluid-flow along potentially active faults in
crystalline rock. Geology, 23, 8: 683–686
Barton N., S. Bandis, K. Bakhtar (1985). Strength, deformation and conductivity coupling of
rock joints. Int. J. Rock Mech. Min. Sic. & Geomech. Abstr. 22, 121-140.
Barton, C.A., S. Hickman, R. Morin, M.D. Zoback, R. Benoit (1998). Reservoir-scale fracture
permeability in the Dixie Valley, Nevada, Geothermal Field, paper 47371 presented at
SPE/ISRM Eurock '98, Soc. of Pet. Eng., Trondheim, Norway.
Barton, N.A., Choubey, V., 1977. The shear strength of rock joints in theory and practice. Rock
Mechanics 10, 1-34.
Batchelor, A. S., R. Baria, and K. Hearn (1983), Monitoring the effects of hydraulic stimulation
by microseismic event location: a case study, in *58th Ann. Tech. Conf. and Exhibition of SPE*,
edited, Soc. Petrol. Eng., San Francisco, California.
Baujard, C., Bruel, D., (2006). Numerical study of the impact of fluid density on the pressure
distribution and stimulated volume in the Soultz HDR reservoir, Geothermics, 35, 607–
621.Bennour Z, Ishida T, Nagaya Y, Chen Y, Nara Y, Chen Q, Sekine K, Nagano Y (2015)
Crack extension in hydraulic fracturing of shale cores using viscous oil, water, and liquid
carbon dioxide. Rock Mech Rock Eng 48(4):1463–1473Biot, M.A. (1941). General theory of
three dimensional consolidation. Journal of Applied Physics. 12: 155–164.
Blake, K., and N. Davatzes (2011), Crustal stress heterogeneity in the vicinity of COCO
geothermal field, CA., paper presented at 36th Workshop on Geothermal Reservoir
Engineering, Stanford University, Stanford University, Jan31-Feb2.
Block L., Wood C., Yeck W., King V. (2015). Induced seismicity constraints on subsurface
geological structure, Paradox Valley, Colorado. Geophysical Journal International, 200(2),
1170–1193. doi:10.1093/gji/ggu459
Bommer, J.J., Oates, S., Cepeda, J.M., Lindholm, C., Bird, J., Torres, R., Marroquʹın, G., Rivas,
J., (2006). Control of hazard due to seismicity induced by a hot fractured rock geothermal
project, Eng. Geol., 83, 287–306.
Boroumand N, Eaton D. (2012). Comparing Energy Calculations - Hydraulic Fracturing and
Microseismic Monitoring. Presented at the Geoconvention 2012 - 74th Mtg., EAGE,
Copenhagen, C042.
Breede, K., Dzebisashvili, K., Liu, X., and Falcone, G. (2013). A systematic review of enhanced
(or engineered) geothermal systems: past, present and future. Geothermal Energy, 1(1):1.
Brown, D.W., (2000). A hot dry rock geothermal energy concept utilizing supercritical CO2
instead of water. In: Proceedings of the Twenty-Fifth Workshop on Geothermal Reservoir
Engineering, Stanford, CA. Stanford University.
Brown, D. W., Duchane, D. V., Heiken, G., and Hriscu, V. T. (2012). Mining the Earth's heat:
hot dry rock geothermal energy. Springer Science & Business Media.
Bruno M, Nakagawa F. (1991). Pore pressure influence on tensile fracture propagation in
sedimentary rock. International Journal of Rock Mechanics and Mining Sciences &
Geomechanics Abstracts, 28(4), 261–273. doi:10.1016/0148-9062(91)90593-b
Bunger A, Detournay E, Garagash D, Peirce A, others. (2007). Numerical simulation of
hydraulic fracturing in the viscosity dominated regime. In SPE Hydraulic Fracturing
Technology Conference. Society of Petroleum Engineers.
Bunger AP, Jeffrey RG, Kear J, Zhang X. (2011). Experimental Investigation of the Interaction
among Closely Spaced Hydraulic Fractures. In 45th US Rock Mechanics / Geomechanics
Symposium (pp. 11–318+). San Francisco.
Buscheck, T.A., J.M. Bielicki, T.A. Edmunds, Y. Hao, Y. Sun, J.B. Randolph, and M.O. Saar
(2016). Multifluid geo-energy systems: Using geologic $CO_2$ storage for geothermal energy
production and grid-scale energy storage in sedimentary basins, Geosphere, DOI:
10.1130/GES01207.1, 12(3):678-696.
Byerlee, J. (1978). Friction of rocks. Pure and applied geophysics, 116(4-5):615-626.
Caine, J. S., Evans, J. P., and Forster, C. B. (1996). Fault zone architecture and permeability
structure. Geology, 24(11):1025-1028.
~~Calo et al. (2011) Valentin~~

Calò, M., Dorbath, C., Cornet, F. H., & Cuenot, N. (2011). Large-scale aseismic motion identified through 4-DP-wave tomography. Geophysical Journal International, 186(3), 1295-1314.

Candela, T., Brodsky, E. E., Marone, C., & Elsworth, D. (2014). Laboratory evidence for particle mobilization as a mechanism for permeability enhancement via dynamic stressing. Earth and Planetary Science Letters, 392, 279-291.

Catalli, F., M.-A. Meier, S. Wiemer (2013). Coulomb stress changes at the Basel geothermal site: can the Coulomb model explain induced seismicity in an EGS? Geophys. Res. Let., 40.

Chacón E, Barrera V, Jeffrey R, van As A. (2004). Hydraulic fracturing used to precondition ore and reduce fragment size for block caving. Presented at the MassMin 2004 Santiago Chile.

Chen, Z., S. P. Narayan, Z. Yang, and S. S. Rahman (2000), An experimental investigation of hydraulic behaviour of fractures and joints in granitic rock, *Int. J. Rock Mech. & Min. Sci.*, *37*, 1061-1071.

Chitrala, Y., C. Moreno, C. H. Sondergeld, and C. S. Rai (2010), Microseismic mapping of laboratory induced hydraulic fractures in anisotropic reservoirs, paper presented at Tight Gas Completions Conference, Society of Petroleum Engineers.

Cornet F. H., Helm J., Poitrenaud H., Etchecopar A. (1997). Seismic and Aseismic Slips Induced by Large-scale Fluid Injections. Pure appl. geophys. 150 (1997) 563–583

Cornet F.H., Li L., Hulin J.-P., Ippolito I., Kurowski P. (2003). The hydromechanical behaviour of a fracture: an in situ experimental case study. International Journal of Rock Mechanics & Mining Sciences 40 (2003) 1257–1270

Cornet, F. H., and J. Desroches (1989), The problem of channeling in Hot Dry Rock reservoirs, paper presented at Camborne School of Mines Intenational Hot Dry Rock Conference, Robertson Scientific Publishers, Llandudno, UK, Cornwall, UK.

Cornet, F. H., and O. Scotti (1993), Analysis of induced seismicity for fault zone identification, *Int. J. Rock Mech. Min. Sci. & Geomech. Abst.*, *30*(7), 789-795.

Cornet, F. H., and R. H. Jones (1994), Field evidence on the orientation of forced water flow with respect to the regional principal stress directions, paper presented at 1st North American Rock Mechanics Symposium, Balkema, Austin, Texas.

Cornet, F.H. (2012). The relationship between seismic and aseismic motions induced by forced fluid injections. Hydrogeology Journal 20: 1463–1466.

Das I, Zoback MD. (2011). Long-period, long-duration seismic events during hydraulic fracture stimulation of a shale gas reservoir. The Leading Edge, 30(7), 778–786. doi:10.1190/1.3609093

Davies, R., Foulger, G., Bindley, A., Styles, P. (2013). Induced seismicity and hydraulic fracturing for the recovery of hydrocarbons, Marine and Petroleum Geology, 45, 171-185.

Deichmann, N., J. Ernst (2009). Earthquake focal mechanisms of the induced seismicity in 2006 and 2007 below Basel (Switzerland), Swiss J Geosci, 102(3), 457-466.

Deichmann, N., Kraft, T., Evans, K.F, (2014). Identification of faults activated during the stimulation of the Basel geothermal project from cluster analysis and focal mechanisms of the larger magnitude events. Geothermics, 52 (2014) 84–97.

Derode B., F. Cappa, Y. Guglielmi, J. Rutqvist (2013). Coupled seismo-hydromechanical monitoring of inelastic effects on injection-induced fracture permeability. International Journal of Rock Mechanics & Mining Sciences 61: 266–274

Detournay, E. (2016). Mechanics of Hydraulic Fractures. In Davis, SH and Moin, P, (eds), Annual Review of Fluid Mechanics, vol 48, p. 311-339.

Dusseault MB, McLennan J, Shu J. (2011). Massive multi-stage hydraulic fracturing for oil and gas recovery from low mobility reservoirs in China. Petroleum Drilling Techniques, 39(3), 6–16.

Edwards, B., Kraft, T., Cauzzi, C., Kaestli, P., and Wiemer, S. (2015). Seismic monitoring and analysis of deep geothermal projects in St Gallen and Basel, Switzerland. Geophysical Journal International, 201(2):1020-1037.

Ellsworth, W.L. (2013). Injection-induced earthquakes. Science, 12, 341, 6142

Elsworth, D., Fang, Y., Gan, Q., Im, K. J., Ishibashi, T., & Guglielmi, Y. (2016, April). Induced seismicity in the development of EGS—benefits and drawbacks. In Rock Dynamics: From Research to Engineering: Proceedings of the 2nd International Conference on Rock Dynamics and Applications (p. 13). CRC Press.

Emmermann R, Lauterjung J. (1997). The German Continental Deep Drilling Program KTB: Overview and major results. J. Geophys. Res., 102(B8), 18179–18201. doi:10.1029/96jb03945

Esaki T., H. Hojo, T. Kimura, N. Kameda, E. (1991). Shear-Flow Coupling Test on Rock joints. Proceedings – Seventh International Congress on Rock Mechanics, Vol 1: Rock Mechanics and Environmental Protection.

Esaki, T., Du, S., Mitani, Y., Ikusada, K., Jing, L., (1999). Development of a shear-flow test
apparatus and determination of coupled properties for a single rock joint, Int. J. Rock Mech.
Min. Sci., 36, 641–650.
Evans K. F., F. H. Cornet, T. Hashida, K. Hayashi, T. Ito, K. Matsuki, T. Wallroth (1999).
Stress and rock mechanics issues of relevance to HDR/HWR engineered geothermal systems:
review of developments during the past 15 years. Geothermics 28, 455-474
Evans K. F., H. Moriya, H. Niitsuma, R.H. Jones, W.S. Phillips, A. Genter, J. Sausse, R. Jung,
R. Baria (2005a). Microseismicity and permeability enhancement of hydrogeologic structures
during massive fluid injections into granite at 3 km depth at the Soultz HDR site, Geophys. J.
Int., 160, 388–412.
Evans K.F. (2005). Permeability creation and damage due to massive fluid injections into
granite at 3.5 km at Soultz: 2. Critical stress and fracture strength, J. geophys. Res., 110.
Evans K.F., A. Genter, J. Sausse (2005b). Permeability creation and damage due to massive
fluid injections into granite at 3.5 km at Soultz: 1. Borehole observations. J. geophys. Res., 110,
B04203.
Evans K.F., F. Wyatt (1984). Water table effects on the measurement of earth strain.
Tectonophysics, 108: 323-337
Evans K.F., T. Kohl, L. Rybach, R.J. Hopkirk (1992). The effect of fracture normal compliance
on the long-term circulation behaviour of a hot dry rock reservoir: A parameter study using the
new fully coupled code fracture. Geothermal Resources Council Transactions, Vol. 16, 449-
456, San Diego, CA
Evans, J.P., Forster, C.B., Goddard, J.V. (1997). Permeability of fault-related rocks, and
implications for hydraulic structure of fault zones. J. Struct. Geol. 19, 1393–1404.
Evans, K. F. (1983), Some examples and implications of observed elastic deformations
associated with the growth of hydraulic fractures in the Earth, paper presented at Workshop on
Hydraulic Fracturing Stress Measurements, National Academy Press, Monterey, California.
Evans, K. F. (1998). Does significant aseismic slip occur on fractures in HDR systems under
stimulation conditions? Proceedings, 4th Int. HDR Forum Strasbourg, September 28-30th.
Evans, K. F., and P. Meier (1995), Hydro-jacking and hydrofracturing tests in a fissile schist in
south-west Switzerland: In-situ stress characterisation in difficult rock, paper presented at 2nd
Int. Conf. on the Mechanics of Jointed and Faulted Rock, Balkema, Vienna, 10-14 April.
Evans, K. F., and S. Sikaneta (2013), Characterisation of natural fractures and stress in the Basel
reservoir from wellbore observations (Module 1), in *GEOTHERM: Geothermal Reservoir*
*Processes: Research towards the creation and sustainable use of Enhanced Geothermal*
*Systems*, edited by K. F. Evans, pp. 9-18, Swiss Federal Office of Energy Publication No
290900, Bern.
Evans, K. F., Zappone, A., Kraft, T., Deichmann, N., and Moia, F. (2012). A survey of the
induced seismic responses to uid injection in geothermal and co 2 reservoirs in Europe.
Geothermics, 41: 30-54.
Evans, K., Holzhausen, G. (1983). On the development of shallow hydraulic fractures as viewed
through the surface deformation field: Part 2-case histories. Journal of Petroleum Technology,
1207  35(02):411-420.

Evans, K., Wieland, U., Wiemer, S., Giardini D. (2014). Deep Geothermal Energy R&D
Roadmap for Switzerland, 2014.
Evans, K. F. (2014), Reservoir Creation, in Energy from the Earth - Deep Geothermal as a
Resource for the Future?, edited by S. Hirschberg, S. Wiemer and P. Burgherr, pp. 82-118,
Zentrum für Technologiefolgen-Abschätzung, Bern.
Eyre, T. S., and M. van der Baan (2015), Overview of moment-tensor inversion of microseismic
events, *The Leading Edge*, *August*, 882-888 doi: 10.1190/tle34080882.1.
Fang, Y., Elsworth, D., Wang, C., Ishibashi, T., & Fitts, J. P. (2017). Frictional stability-
permeability relationships for fractures in shales. Journal of Geophysical Research: Solid Earth,
1217  122(3), 1760-1776.

Fang, Y., Elsworth, D., & Cladouhos, T. T. (2018). Reservoir permeability mapping using
microearthquake data. Geothermics, 72, 83-100.
Faulkner D., Jackson, C., Lunn, R., Schlische, R., Shipton, Z., Wibberley, C., Withjack, M.,
(2010). A review of recent developments concerning the structure, mechanics and fluid flow
properties of fault zones. J. Struct. Geol. 32, 1557–1575.
Faulkner D.R., and E.H. Rutter (2008). Can the maintenance of overpressured fluids in large
strike-slip fault zones explain their apparent weakness? Geology 29, no. 6: 503–506.
Gale, J. E. (1975). A numerical, field and laboratory study of flow in rocks with deformable
fractures. Ph.D. dissertation, Berkeley, University of California, 255 p.

Gale, J. E. (1993). Fracture properties from laboratory and large scale field tests: evidence of scale effects. Scale Effects in Rock Masses. Proc. 2nd Int. Workshop on Scale Effects in Rock Masses (Edited by Pinto da Cunha A.), Lisbon, pp. 341-352. Balkema, Rotterdam.

Garagash, D.I., L.N., Germanovich (2012). Nucleation and arrest of dynamic slip on a pressurized fault, J. Geophys. Res., 117, B10310.

Garapati, N., J.B. Randolph, and M.O. Saar (2015). Brine displacement by $CO_2$, energy extraction rates, and lifespan of a $CO_2$-limited $CO_2$ Plume Geothermal (CPG) system with a horizontal production well, Geothermics, DOI: 10.1016/j.geothermics.2015.02.005, 55:182–194.

Garcia, J., Walters, M., Beall, J., Hartline, C., Pingol, A., Pistone, S., & Wright, M. (2012, January). Overview of the northwest Geysers EGS demonstration project. In Proceedings, Thirty-Seventh Workshop on Geothermal Reservoir Engineering Stanford University.

Genter A. , Goerke X., Graff J.-J, Cuenot N., Krall G., Schindler M., Ravier G. (2010). Current Status of the EGS Soultz Geothermal Project (France). Proceedings World Geothermal Congress 2010, Bali, Indonesia, 25-29 April 2010

Gentier, S., D. Hopkins, and J. Riss (2000). Role of fracture geometry in the evolution of flow paths under stress, in Dynamics of Fluids in Fractured Rock, Geophys. Monogr. Ser., vol. 122, edited by B. Faybishenko, P. A. Witherspoon, and S. M. Benson, pp. 169 – 184, AGU, Washington, D. C.

Giardini, D. (2009). Geothermal quake risks must be faced, Nature, 462 (7275), 848-849.

Gischig, V., G. Preisig (2015), Hydro-fracturing versus hydro-shearing: a critical assessment of two distinct reservoir stimulation mechanisms, paper presented at International Congress of Rock Mechanics, ISRM 2015, Montréal, Canada.

Gischig, V.S., J. Doetsch, H. Maurer, H. Krietsch, F. Amann, K.F. Evans, M. Nejati, M.R. Jalali, A. Obermann, B. Valley, S. Wiemer, and D. Giardini (2017). On the link between stress field and small-scale hydraulic fracture growth in anisotropic rock derived from mirco-seismicity. Submitted to Solid Earth.

Gischig, V., Jalali, M., Amann, F., Krietsch, H., Klepikova, M., Esposito, S., Broccardo, M., Obermann, A., Mignan, A., Doetsch, J., Madonna, C. (2016). Impact of the ISC Experiment at the Grimsel Test Site-Assessment of Potential Seismic Hazard and Disturbances to Nearby Experiments and KWO Infrastructure. ETH Zurich, https://doi.org/10.3929/ethz-b-000189973.

Gischig, V. S. (2015). Rupture propagation behavior and the largest possible earthquake induced by fuid injection into deep reservoirs. Geophysical Research Letters, 42(18):7420-7428.

Gischig, V. S., Wiemer, S. Alcolea, A. R. (2014). Balancing reservoir creation and seismic hazard in enhanced geothermal systems. Geophysical Journal International. doi: 10.1093/gji/ggu221

Gischig, V.S., Wiemer, S. (2013). A stochastic model for induced seismicity based on non-linear pressure diffusion and irreversible permeability enhancement, Geophys. J. Int., 194(2), 1229–1249.

Goebel, T. H. W., T. W. Becker, D. Schorlemmer, S. Stanchits, C. Sammis, E. Rybacki, G. Dresen (2012). Identifying fault heterogeneity through mapping spatial anomalies in acoustic emission statistics, J. Geophys. Res., 117, B03310.

Goertz-Allmann, B.P., Wiemer, S. (2013). Geomechanical modeling of induced seismicity source parameters and implications for seismic hazard assessment, Geophysics, 78(1), KS25–KS39.

Goertz-Allmann, B.P.,Goertz, A., Wiemer, S. (2011). Stress drop variations of induced earthquakes at the Basel geothermal site, Geophys. Res. Lett. 38(9), L09308.

Goodman R. E. (1974). The mechanical properties of joints. Proceedings of the 3rd Int. Congr. International Society of Rock Mechanics, Denver, Colorado. National Academy of Sciences, Washington, DC, I, 127–140.

Guglielmi Y., F. Cappa, H. Lancon, J. B. Janowczyk, J. Rutqvist, C. F. Tsang, J. S. Y. Wang (2014). ISRM Suggested Method for Step-Rate Injection Method for Fracture In-Situ Properties (SIMFIP): Using a 3-Components Borehole Deformation Sensor. Rock Mech. Rock Eng. 47: 303–311

Guglielmi Y., F. Cappa, J. Rutqvist, C.-F. Tsang, A. Thoraval (2008). Mesoscale characterization of coupled hydromechanical behavior of a fractured-porous slope in response to free water-surface movement. Int. J. Rock. Mech. Min. Sci. 42: 852–878.

Guglielmi, Y., Cappa, F., Avouac, J.-P., Henry, P., and Elsworth, D. (2015). Seismicity triggered by fuid injection induced aseismic slip. Science, 348(6240):1224-1226.

Guglielmi, Y., F. Cappa, J. Rutqvist, C.-F. Tsang, and A. Thoraval (2006), Field and numerical investigations of free-water surface oscillation effects on rock slope hydromechanical

behaviour – consequences for rock slope stability analyses paper presented at GEOPROC 2006: 2nd International Conference on Coupled Thermo-hydro-mechanicalchemical

Guglielmi, Y.G. and Henry, P. Nussbaum, C. Dick, P. Gout, C. Amann, F. (2015). Underground Research Laboratories for conducting fault activation experiments in shales. 49th US Rock Mechanics / Geomechanics Symposium held in San Francisco, CA, USA, ARMA 15-0489

Gupta, H. K. (1992), *Reservoir-induced Earthquakes*, 364 pp., Elsevier, Amsterdam, The Netherlands.

Haimson, B. C., and F. H. Cornet (2003), ISRM suggested methods for rock stress estimation-Part 3: hydraulic fracture (HF) and/or hydraulic testing of pre-existing fractures (HTPF), *Int. J. Rock Mech. Min. Sci.*, *40*, 1011-1020.

Hampton, J. C., L. Matzar, D. Hu, and M. Gutierrez (2015), Fracture dimension investigation of laboratory hydraulic fracture interaction with natural discontinuity using acoustic emission, paper presented at 49th US Rock Mechanics/Geomechanics Symposium, Americal Rock Mechanics Association, San Francisco, 28 June-1 July.

Häring, M.O., Schanz, U., Ladner, F. & Dyer, B.C., (2008). Characterization of the Basel 1 enhanced geothermal system, Geothermics, 37, 469–495.

Healy, J. H., W. W. Rubey, D. T. Griggs, and C. B. Raleigh (1968), The Denver earthquakes, *Science*, *161*, 1301-1310.

Hickman, S. H., M. Zoback, C. A. Barton, R. Benoit, J. Svitek, Summer, and R. (2000), Stress and permeability heterogeneity within the Dixie Valley geothermal reservoir: recent results from well 82-5, paper presented at Twenty-Fifth Workshop on Geothermal Reservoir Engineering, Stanford University, Stanford University, Stanford, CA, Jan 24-26.

Hickman, S., M. D. Zoback, and R. Benoit (1998), Tectonic controls on fault-zone permeability in a geothermal reservoir at Dixie Valley, Nevada, paper 47213 presented at SPE/ISRM Eurock '98, Soc. of Pet. Eng., Trondheim, Norway.

Hogarth, R., H. Holl, and A. McMahon (2013), Flow testing results from Habanero EGS Project, paper presented at Australian Geothermal Energy Conferences, Brisbane, Australia, 14-15 November.

Horálek, J., Z. Jechumtálová, L. Dorbath, and J. Síleny (2010), Source mechanisms of micro-earthquakes induced in a fluid injection experiment at the HDR site Soultz-sous-Forˆets (Alsace) in 2003 and their temporal and spatial variations, *Geophys. J. Int.*, *181*, 1547-1565 doi: 10.1111/j.1365-246X.2010.04506.x.

Houben, G. (2015), Review: Hydraulics of water wells—flow laws and influence of geometry, *Hydrogeology J.*, *23*, 1633-1657.

Hubbert, M. K. and Rubey, W. W. (1959). Role of uid pressure in mechanics of overthrust faulting i. mechanics of uid-_lled porous solids and its application to overthrust faulting. Geological Society of America Bulletin, 70(2):115{166.

Hummel, N., and T. M. Müller (2009), Microseismic signatures of non-linear pore-fluid pressure diffusion, Geophys. J. Int., 179, 1558-1565 doi: 10.1111/j.1365-246X.2009.04373.x.

Husen, S., C. Bachmann, D. Giardini (2007). Locally triggered seismicity in the central Swiss Alps following the large rainfall event of August 2005. Geophysical Journal International, 171 (2007), pp. 1126-1134, 10.1111/j.1365-246X.2007.03561.x

Huw, C., Eisner, L., Styles, P., Turner, P., (2014). Felt seismicity associated with shale gas hydraulic fracturing: The first documented example in Europe, Geophysical Research Letter, doi: 10.1002/2014GL062047

Ishida T. (2001). Acoustic emission monitoring of hydraulic fracturing in laboratory and field. Construction and Building Materials 15 Ž2001. 283-295

Jaeger JC. (1963). Extension Failures in Rocks subject to fluid Pressure. Journal of Geophysical Research, 68(21), 6066–6067.

Jeanne, P., Rutqvist, J., Rinaldi, A. P., Dobson, P. F., Walters, M., Hartline, C., & Garcia, J. (2015). Seismic and aseismic deformations and impact on reservoir permeability: The case of EGS stimulation at The Geysers, California, USA. Journal of Geophysical Research: Solid Earth, 120(11), 7863-7882.

Jeanne, P., Rutqvist, J., Vasco, D., Garcia, J., Dobson, P. F., Walters, M., Hartline, C., Borgia, A. (2014). A 3D hydrogeological and geomechanical model of an Enhanced Geothermal System at The Geysers, California. Geothermics, 51, 240-252.

Jeanne, P., Y. Guglielmi, and F. Cappa. 2012. Dissimilar properties within a carbonate-reservoir's small fault zone, and their impact on the pressurization and leakage associated with CO2 injection. Journal of Structural Geology. DOI:10.1016/j.jsg.2012.10.010

Jeffrey R, Enever J, Phillips R, Ferguson T, Davidson S, Bride J. (1993). Small-Scale Hydraulic Fracturing and Mineback Experiment in Coal Seams. Presented at the Proceedings of the 1993 International Coalbed methane Symposium.

Jeffrey RG, Brynes RP, Lynch PJ, Ling DJ. (1992). An Analysis of Hydraulic Fracture and
Mineback Data for a Treatment in the German Creek Coal Seam. Society of Petroleum
Engineers. doi:10.2118/24362-MS
Jeffrey RG, Bunger A. (2007). A Detailed Comparison of Experimental and Numerical Data
on Hydraulic Fracture Height Growth Through Stress Contrasts. Society of Petroleum
Engineers. doi:10.2118/106030-MS
Jeffrey RG, Bunger AP, Lecampion B, Zhang X, Chen ZR, van As A, et al. (2009). Measuring
Hydraulic Fracture Growth in Naturally Fractured Rock. In 2009 SPE Annual Technical
Conference and Exhibition (p. SPE 124919+). New Orleans, Louisiana, USA: SPE.
Jeffrey RG, Settari A. (1995). A Comparison of Hydraulic Fracture Field Experiments,
Including Mineback Geometry Data, with Numerical Fracture Model Simulations. Society of
Petroleum Engineers. doi:10.2118/30508-MS
Johnson E, Cleary MP. (1991). Implications of recent laboratory experimental results for
hydraulic fractures. Society of Petroleum Engineers. doi:10.2118/21846-MS
Jost M, Büßelberg T, Jost Ö, Harjes H. (1998). Source parameters of injection-induced
microearthquakes at 9 km depth at the KTB Deep Drilling site, Germany. Bulletin of the
Seismological Society of America, 88(3), 815–832.
Jung R. (1989). Hydraulic in situ investigations of an artificial fracture in the Falkenberg
Granite. Int. J. Rock Mech. Min. Sci. & Geomech. Abstr. 26: 301-308.
Jung R. (2013). EGS — Goodbye or Back to the Future. Effective and Sustainable Hydraulic
Fracturing, http://dx.doi.org/10.5772/56458
Jupe A. Green A. S. P., Wallroth T. (1992). Induced Microseismicity and Reservoir Growth at
the Fjällbacka Hot Dry Rocks Project, Sweden. Int. J. Rock Mech. Min. Sci. & Geomech.
Abstr. Vol. 29. No. 4. pp. 343-354.
Kaieda, H., Jones, R., Moriya, H., Sasaki, S. & Ushijima, K., (2005). Ogachi HDR reservoir
evaluation by AE and geophysical methods, in Proceedings of World Geothermal Congress
2005, Antalya, Turkey, April 24–29.
Karvounis, D.C., Gischig, V.S., Wiemer, S., (2014). Towards a Real-Time Forecast of Induced
Seismicity for Enhanced Geothermal Systems. Proceedings of the 2014 Shale Energy
Engineering Conference, July 21–23, 2014, Pittsburgh, Pennsylvania, 246.
Keranen, K., M, Savage, H. M., Abers, G. A., & Cochran, E. S. (2013). Potentially induced
earthquakes in Oklahoma, USA: Links between wastewater injection and the 2011 Mw 5.7
earthquake sequence, Geology 41 (6), 699–702, doi:10.1130/G34045.1
Keusen, H.R., Ganguin, J., Schuler, P., Buletti, M., 1989. Grimsel test site: Geology. Nationale
Genossenschaft fuer die Lagerung Radioaktiver Abfaelle (NAGRA), Baden, Switzerland.
Technical Report NTB 87-14E, 166 pp.
Király, E., Zechar, J.D., Gischig, V.S, Karvounis, D., Doetsch, J., Wiemer, S., (2015). Modeling
Induced Seismicity and Validating Models in Deep Geothermal Energy Projects. In
preparation.
Kohl, T., K. F. Evans, R. J. Hopkirk, R. Jung, and L. Rybach (1997), Observation and
simulation of non-Darcian flow transients in fractured rock, *Wat. Resourc. Res.*, *33*(3), 407-

1392 418.

Krietsch, H., V. Gischig, R. Jalali, F. Amann, K. F. Evans, J. Doetsch, and B. Valley (2017),
Stress measurements in crystalline rock: Comparison of overcoring, hydraulic fracturing and
induced seismicity results, in *ARMA 51st US Rock Mechanics / Geomechanics Symposium*,
edited, San Francisco, California, USA.
Lee, H. S., and T. F. Cho (2002), Hydraulic Characteristics of Rough Fractures in Linear Flow
under Normal and Shear Load, *Rock Mech. Rock Eng.*, *35*, 299-318 DOI 10.1007/s00603-002-
0028-y.
Lee, H.S., Cho, T.F. (2002). Hydraulic characteristics of rough fractures in linear flow under
normal and shear load, Rock Mech. Rock Eng., 35(4), 299–318.
Louis, C., J.-L., Dessene, B. Feuga (1977). Interaction between water flow phenomena and the
mechanical behavior of soil or rock masses. Gudehns, G., ed., Finite elements in geomechanics:
New York, John Wiley S Sons, 572 p.
Majer, E., J. Nelson, A. Robertson-Tait, J. Savy, and I. Wong (2012). Protocol for addressing
induced seismicity associated with enhanced geothermal systems, U.S. Department of Energy,
Energy Efficiency and Renewable Energy.
Manning, C. and Ingebritsen, S. (1999). Permeability of the continental crust: Implications of
geothermal data and metamorphic systems. Reviews of Geophysics, 37(1):127/150.
Marone, C., and C. H. Scholz (1988), The depth of seismic faulting and the upper transition
from stable to unstable slip regimes, *Geophys. Res. Lett.*, *15*(6), 621-624 DOI:
10.1029/GL015i006p00621.

Martínez-Garzón, P., Kwiatek, G., Bohnhoff, M., & Dresen, G. (2017). Volumetric components in the earthquake source related to fluid injection and stress state. Geophysical Research Letters, 44(2), 800-809.

Martínez-Garzón, P., Bohnhoff, M., Kwiatek, G., & Dresen, G. (2013). Stress tensor changes related to fluid injection at The Geysers geothermal field, California. Geophysical Research Letters, 40(11), 2596-2601.

Martin C. D., C. C. Davison, E. T. Kozak (1990). Characterizing normal stiffness and hydraulic conductivity of a major shear zone in granite. Rock joints, eds. Barton & Stephansson, Balkema, Rotterdam, Netherlands.

Maury, V., 1994. Rock failure mechanisms identification: A key for wellbore stability and reservoir behaviour problem, in Eurock 94, edited by Delft, Netherlands, 29-31 August, 175-182, Balkema.

Maxwell, S. (2014), Microseismic Imaging of Hydraulic Fracturing: Improved Engineering of Unconventional Shale Reservoirs, 197 pp., Society of Exploration Geophysicists.

McClure, M. W. (2015), Generation of large postinjection-induced seismic events by backflow from dead-end faults and fractures, Geophysical Research Letters, 42(6647–6654).

McClure M.W., R. N. Horne (2011). Investigation of injection-induced seismicity using a coupled fluid flow and rate/state friction model. Geophysics 76, 6.

McClure M.W., R. N. Horne (2014). An investigation of stimulation mechanisms in Enhanced Geothermal Systems International Journal of Rock Mechanics & Mining Sciences 72: 242–260

McClure, M. W. (2012). Modeling and characterization of hydraulic stimulation and induced seismicity in geothermal and shale gas reservoirs. PhD thesis, Stanford University.

McGarr, A. (1976). Seismic moments and volume changes, Journal of Geophysical Research, 81(8), 1487-1494.

Mena, B.,Wiemer, S., Bachmann, C., (2013). Building robust model to forecast the induced seismicity related to geothermal reservoir enhancements, Bull. seism. Soc. Am., 103(1), 383–393.

Meng, C. (2011), Hydraulic fracture propagation in pre-fractured rocks, paper presented at SPE Hydraulic Fracturing Technology Conference and Exhebition, SPE, The Woodlands, Texas, 24-26 Jan.

Mignan, A., Landtwing, D., Kästli, P., Mena, B., Wiemer, S. (2015). Induced seismicity risk
analysis of the 2006 Basel, Switzerland, Enhanced Geothermal System project: Influence of
uncertainties on risk mitigation, Geothermics, 53 (2015) 133–146.
Murdoch LC, Schweisinger T, Svenson E, Germanovich L. (2004). Measuring and analyzing
transient changes in fracture aperture during well tests: preliminary results. In: Dynamics of
fluids in fractured rock (Witherspoon Conference). LBL Report 54275, February 10–14, 2004.
p. 129–32.
Murphy H., C. Huang, Z. Dash, G. Zyvoloski, A. White (2004). Semi-analytical solutions for
fluid flow in rock joints with pressure-dependent openings. Water Resources Research 40,
W12506
Nicholson, C., and R. L. Wesson (1990), Earthquake Hazard Associated with Deep Well
Injection-A Report to the U.S. Environmental Protection Agency, 1951, US Geological Survey
Bulletin.
Nicol, D. A. C., and B. A. Robinson (1990), Modelling the heat extraction from the
Rosemanowes HDR reservoir, *Geothermics*, *19*, 247-257.
Niitsuma H., M. Fehler, R. Jones, S. Wilson, J. Albright, A. Green, R. Baria, K. Hayashi, H.
Kaieda, K. Tezuka, A. Jupe, T. Wallroth, F. Cornet, H. Asanuma, H. Moriya, K. Nagano, W.S.
Phillips, J. Rutledge, L. House, A. Beauce, D. Alde, R. Aster (1999). Current status of seismic
and borehole measurements for HDR/HWR development. Geothermics, 28, 4-5: 475-490.
Olsson R., N. Barton (2001). An improved model for hydromechanical coupling during
shearing of rock joints. International Journal of Rock Mechanics and Mining Sciences, 38, 3:
317–329.
Parker R. (1999). The Rosemanowes HDR project 1983-1991. Geothermics, 28, 603-615.
Parker, R. H. (1989a), Hot Dry Rock Geothermal Energy: Phase 2B Final Report of the
Camborne School of Mines Project, 1391 pp., Pergamon Press, Oxford.
Pearson, C. (1981), The relationship between microseismicity and high pore pressures during
hydraulic stimulation experiments in low porosity granitic rock, *J. Geophys. Res.*, *86*, 7855-
1470 7864.

Pettitt W, Pierce M, Damjanac B, Hazzard J, Lorig L, Fairhurst C, et al. (2011). Fracture
network engineering for hydraulic fracturing. The Leading Edge, 30(8), 844–853.
doi:10.1190/1.3626490

Petty, S., Nordin, Y., Glassely, W., Cladouhos, T. (2013). Improving geothermal project economics with multi-zone stimulation: results from the Newberry volcano EGS demonstration. Proc. 38th Works. Geoth. Rese. Eng., Stanford University, SGP-TR-198.

Phillips, S., L. S. House, and M. C. Fehler (1997), Detailed joint structure in a geothermal reservoir from studies of induced microseismic clusters, *J. Geophys. Res.*, *102*(B6), 11,745-711,763.

Pine, R.J., Baria, R., Pearson, R.A., Kwakwa, K., McCartney, R (1987). A Technical Summary of Phase 2B of the Camborne School of Mines HDR Project, 1983-1986. Geothermics, 16, 4: 341-353.

Potter, R., Robinson, E., and Smith, M. (1974). Method of extracting heat from dry geothermal reservoirs. US Patent 3,786,858.

Power, W. L., and T. E. Tullis (1991), Euclidean and fractal models for the description of surface roughness, *J. Geophys. Res.*, *96*(B1), 415-424.

Preisig, G., E. Eberhardt, V. Gischig, V. Roche, M. Van der Baan, B. Valley, P. Kaiser, and D. Du (2015), Development of connected rock mass permeability by hydraulic fractures growth accompanying fluid injection, Geofluids 15, 321–337. Rahman, M.K., Hossain, M.M., Rahman, S.S. (2002). A shear-dilation-based model for evaluation of hydraulically stimulated naturally fractured reservoirs. International Journal for Numerical and Analytical Methods in Geomechanics, 26, 5: 469-497.

Pruess, K., (2006). Enhanced geothermal systems (EGS) using $CO_2$ as working fluid – a novel approach for generating renewable energy with simultaneous sequestration of carbon. Geothermics 35 (4), 351–367.

Pruess, K., (2007). Role of fluid pressure in the production behavior of enhanced geothermal systems with $CO_2$ as working fluid. GRC Trans. 31, 307–311.

Raleigh, C., Healy, J., and Bredehoeft, J. (1976). An experiment in earthquake control at rangely, colorado. work (Fig. Ib), 108(52):30.

Randolph, J.B., and M.O. Saar (2011a), Combining geothermal energy capture with geologic carbon dioxide sequestration, Geophysical Research Letters, DOI: 10.1029/2011GL047265, 38, L10401.

Randolph, J.B. and M.O. Saar (2011b). Coupling carbon dioxide sequestration with geothermal
energy capture in naturally permeable, porous geologic formations: Implications for $CO_2$
sequestration, Energy Procedia, DOI: 10.1016/j.egypro.2011.02.108, 4:2206-2213.
Rutledge, J. T., Phillips, W. S., & Mayerhofer, M. J.: Faulting Induced by Forced Fluid
Injection and Fluid Flow Forced by Faulting: An Interpretation of Hydraulic-Fracture
Microseismicity, Carthage Cotton Valley Gas Field, Texas, Bulletin of the Seismological
Society of America, 94, (2004),1817.
Rutqvist J. (1995): Determination of hydraulic normal stiffness of fractures in hard rock from
well testing. Int. J. Rock Mech. Min. Sci.1, 32: 513–23.
Rutqvist J., O. Stephansson (2003). The role of hydromechanical coupling in fractured rock
engineering. Hydrogeology Journal, 11, 1:7–40.
Rutqvist, J. (2011). Status of the tough-ac simulator and recent applications related to coupled
fluid flow and crustal deformations. Computers & Geosciences, 37(6):739-750.
Rutqvist, J., and C. M. Oldenburg (2008), Analysis of injection-induced micro-earthquakes in
a geothermal stream reservoir, Geysers Geothermal Field, California, Proceedings of the 42th
U. S. Rock Mechanics Symposium, June 29–July 2, 2008, San Francisco, California, USA, 151.
Rutqvist, J., and O. Stephansson (1996), A cyclic hydraulic jacking test to determine the in-situ
stress normal to a fracture, *Int. J. Rock Mech. Min. Sci. & Geomech. Abstr.*, *33*(7), 695-711.
Saar, M.O., and M. Manga (2003). Seismicity induced by seasonal groundwater recharge at Mt.
Hood, Oregon, Earth and Planetary Science Letters, DOI: 10.1016/S0012-821X(03)00418-7,
1523 214:605-618.

Saar, M.O., and M. Manga (2004). Depth dependence of permeability in the Oregon Cascades
inferred from  hydrogeologic, thermal, seismic, and magmatic modeling constraints, Journal of
Geophysical   Research, DOI: 10.1029/2003JB002855, 109, Nr. B4, B04204.
Saar, M.O. (2011). Review: Geothermal heat as a tracer of large-scale groundwater flow and
as a means to determine permeability fields, Hydrogeology Journal, DOI: 10.1007/s10040-010-
1529 0657-2, 19:31-52, 2011.

Saar, M.O. (to be published 2017). Novel Geothermal Technologies, in Potentials, costs and
environmental assessment of electricity generation technologies, edited by C. Bauer and S.
Hirschberg, Swiss Federal Office of Energy, Swiss Competences Center for Energy Research
"Supply of Electricity", Swiss Competence Center for Energy Research "Biomass for Swiss
Energy Future".
Samuelson, J., Elsworth, D., & Marone, C. (2009). Shear-induced dilatancy of fluid saturated
faults: Experiment and theory. Journal of Geophysical Research: Solid Earth, 114(B12).
Schanz U, Dyer B, Ladner F, Haering MO. (2007). Microseismic aspects of the Basel 1
geothermal reservoir. In 5th Swiss Geoscience Meeting. Geneva.
Schmittbuhl, J., F. Schmitt, and C. H. Scholz (1995), Scaling invariance of crack surfaces, *J.*
*Geophys. Res.*, *100*(B4), 5953-5973.
Schoenball, M., Baujard, C., Kohl, T., Dorbath, L. (2012). The role of triggering by static stress
transfer during geothermal reservoir stimulation, J. geophys. Res., 117, B09307.
Scholz, C. H. (2015), On the stress dependence of the earthquake b value, *Geophys. Res.Lett.*,
*42*, 1399-1402 doi:10.1002/2014GL062863.
Scholz, C.H., 1990. The mechanics of Earthquakes and Faulting. Cambridge University Press,
Cambridge, UK, p. 39.
Schorlemmer, D., S. Wiemer, Wyss, M., (2005). Variations in earthquake size distribution
across different stress regimes, Nature, 437, 539–542.
Schrauf T. W., Evans D. D. (1986). Laboratory Studies of Gas Flow Through a Single Natural
Fracture WATER RESOURCES RESEARCH, VOL. 22, NO. 7, 1038-1050
Schweisinger T., E.J. Swenson, L.C. Murdoch (2009): Introduction to hydromechanical well
tests in fractured rock aquifers. Groundwater 47, 1:69–79
Schweisinger, T., L.C. Murdoch, and C.O. Huey Jr. (2007). Design of a removable borehole
extensometer. Geotechnical Testing Journal 30, no. 3: 202–211.
Scotti O., Cornet F.H. (1994). In situ evidence for fluid induced aseismic slip events along fault
zones. Int J Rock Mech Min 1:347-358.
Shamir, G., and M. D. Zoback (1992), Stress orientation profile to 3.5 km depth near the San
Andreas fault at Cajon Pass, California, *J. Geophys. Res.*, *97*, 5059-5080.
Sileny, J., D. P. Hill, and F. H. Cornet (2009), Non–double-couple mechanisms of
microearthquakes induced by hydraulic fracturing, *J. Geophys. Res.*, *114*, B08307
doi:10.1029/2008JB005987.
Song I, Suh M, Won K, Haimson B. (2001). A laboratory study of hydraulic fracturing
breakdown pressure in tablerock sandstone. Geosciences Journal, 5(3), 263–271.
doi:10.1007/bf02910309
Spada, M., Tormann, T.,Goebel, T., Wiemer, S. (2013). Generic dependence of the frequency-
size distribution of earthquakes on depth and its relation to the strength profile of the crust,
Geophys. Res. Lett., 40(4), 709–714.
Stein, R. S. (1999). The role of stress transfer in earthquake occurrence. Nature, 402(6762):

1569    605-609.

Tenma N., Yamaguchi T., Zyvoloski G. (2008). The Hijiori Hot Dry Rock test site, Japan
evaluation and optimization of heat extraction from a two-layered reservoir. Geothermics 2008;
37:19–52.
Terakawa, T., Miller, S.A., Deichmann, N. (2012). High fluid pressure and triggered
earthquakes in the enhanced geothermal system in Basel, Switzerland, J. Geophys. Res., 117,
B07305.
Tester, J. W., Anderson, B. J., Batchelor, A., Blackwell, D., DiPippo, R., Drake, E., Garnish,J.,
Livesay, B., Moore, M., Nichols, K., et al. (2006). The future of geothermal energy. Impact of
Enhanced Geothermal Systems (EGS) on the United States in the 21st Century, Massachusetts
Institute of Technology, Cambridge, MA, page 372.
Tezuka, K., and H. Niitsuma (2000), Stress estimated using microseismic clusters and its
relationship to the fracture system of the Hijiori Hot Dry Rock reservoir, *Engineering Geology*,
*56*, 47-62.
Tormann, T., B Enescu, J. Woessner and S. Wiemer (2015). Randomness of megathrust
earthquakes implied by rapid stress recovery after the Japan earthquake, Nature Geoscience 8

1585    (2), 152-158.

Tormann, T., S. Wiemer, A. Mignan (2014). Systematic survey of high-resolution b value
imaging along Californian faults: Inference on asperities, J. Geophys. Res. Solid Earth, 119(3),
2029–2054.
Valley, B., and K. F. Evans (2009), Stress orientation to 5 km depth in the basement below
Basel (Switzerland) from borehole failure analysis, *Swiss J. Earth Sci.*, *102*, 467-480 doi:
10.1007/s00015-009-1335-z.
Valley, B., and K. F. Evans (2010), Stress Heterogeneity in the Granite of the Soultz EGS
Reservoir Inferred from Analysis of Wellbore Failure, paper presented at World Geothermal
Congress, International Geothermal Association, Bali, 25-29 April 2010
van As A, Jeffrey R. (2002). Hydraulic fracture growth in naturally fractured rock: mine
through mapping and analyses. Presented at the NARMS-TAC conference, Toronto, Canada.

van As A, Jeffrey RG. (2000). Caving Induced by Hydraulic Fracturing at Northparkes Mines. Presented at the 4th North American Rock Mechanics Symposium, American Rock Mechanics Association. https://www.onepetro.org/conference-paper/ARMA-2000-0353. Accessed 26 September 2015

van As, A., Jeffrey R., Chacónn E. and Barrera, V. (2004). Preconditioning by hydraulic fracturing for bloc caving in a moderately stressed naturally fractured orebody. Proceeding of the Massmin 2004 conference, Santiago Chile, 22-25 August 2004.

van der Baan M, Eaton D, Dusseault M. (2013). Microseismic Monitoring Developments in Hydraulic Fracture Stimulation. In R Jeffrey (Ed.), Effective and Sustainable Hydraulic Fracturing. InTech. http://www.intechopen.com/books/effective-and-sustainable-hydraulic-fracturing/microseismic-monitoring-developments-in-hydraulic-fracture-stimulation. Accessed 25 September 2015

Vermylen J, Zoback MD. (2011). Hydraulic Fracturing, Microseismic Magnitudes, and Stress Evolution in the Barnett Shale, Texas, USA. Society of Petroleum Engineers. doi:10.2118/140507-MS

Vogler D., Amann F., Bayer P., Elsworth D. (2015). Permeability Evolution in Natural Fractures Subject to Cyclic Loading and Gouge Formation. RMRE, 49(9).

Warpinski N. (2009). Microseismic Monitoring: Inside and Out. Journal of Petroleum Technology, 61(11), 80–85. doi:10.2118/118537-JPT

Warpinski N. (2013). Understanding Hydraulic Fracture Growth, Effectiveness, and Safety Through Microseismic Monitoring. In R Jeffrey (Ed.), Effective and Sustainable Hydraulic Fracturing. InTech. http://www.intechopen.com/books/effective-and-sustainable-hydraulic-fracturing/understanding-hydraulic-fracture-growth-effectiveness-and-safety-through-microseismic-monitoring. Accessed 26 September 2015

Warpinski N., L. W. Teufel (1987): Influence of geologic discontinuities on hydraulic fracture propagation. J. Petrol. Technol. 39: 209–20

Warpinski NR, Clark JA, Schmidt RA, Huddle CW. (1982). Laboratory Investigation on the - Effect of In-Situ Stresses on Hydraulic Fracture Containment. Society of Petroleum Engineers Journal, 22(03), 333–340. doi:10.2118/9834-PA

Warpinski NR, Du J. (2010). Source-Mechanism Studies on Microseismicity Induced by Hydraulic Fracturing. Society of Petroleum Engineers. doi:10.2118/135254-MS

Warpinski NR. (1985). Measurement of Width and Pressure in a Propagating Hydraulic
Fracture. Society of Petroleum Engineers Journal, 25(01), 46–54. doi:10.2118/11648-PA
Warren W. E., Schmith C. W. (1985). In Situ Stress Estimates From Hydraulic Fracturing and
Direct Observation of Crack Orientation. Journal of Geophysical Research, Vol. 9, NO. B8,
1632 829-68

Wehrens, P. (2015). Structural evolution in the Aar Massif (Haslital transect): Implications for
the mid-crustal deformation. PhD thesis, University Bern.
Wolhart, S. L., T. A. Harting, J. E. Dahlem, T. Young, M. J. Mayerhofer, and E. P. Lolon
(2006), Hydraulic fracture diagnostics used to optimize development in the Jonah field, paper
presented at SPE Annual Technical Conference and Exhibition. Society of Petroleum
Engineers.
Yeo I. W., M. H. De Freitas, and R. W. Zimmerman (1998). Effect of shear displacement on
the aperture and permeability of a rock fracture. International Journal of Rock Mechanics and
Mining Sciences, 35, 8:1051–1070
Yoon, J.-S., Zang, A., Stephansson, O., 2014. Numerical investigation on optimized stimulation
of intact and naturally fractured deep geothermal reservoirs using hydro-mechanical coupled
discrete particles joints model. Geothermics, 52.
Zang A, Yoon J.S., Stephansson O, Heidbach O. (2013). Fatigue hydraulic fracturing by cyclic
reservoir treatment enhances permeability and reduces induced seismicity. Geophysical Journal
International, 195(2), 1282–1287. doi:10.1093/gji/ggt301
Zang A., Stephansson O. (2013). Stress Field of the Earth's Crust. Springer Dordrecht
Heidelberg London New York, DOI 10.1007/978-1-4020-8444-7.
Ziegler M, Valley B, Evans K. (2015). Characterisation of Natural Fractures and Fracture Zones
of the Basel EGS Reservoir Inferred from Geophysical Logging of the Basel-1 Well. Presented
at the Proceedings World Geothermal Congress 2015.
Zoback, M. D. and Harjes, H.-P. (1997). Injection-induced earthquakes and crustal stress at 9
km depth at the KTB deep drilling site, Germany. Journal of Geophysical Research: Solid Earth,
102(B8):18477-18491.
Zoback, M. D., Kohli, A., Das, I., McClure, M. W., et al. (2012). The importance of slow slip
on faults during hydraulic fracturing stimulation of shale gas reservoirs. In SPE Americas
Unconventional Resources Conference. Society of Petroleum Engineers.
Zoback, M.D., Rummel, F., Jung R., Raleigh C.B. (1977). Laboratory Hydraulic Fracturing
Experiments in Intact and Pre-fractured Rock Int. J. Rock Mech. Min. Sci. & Geomech. Abstr.
Vol. 14, pp. 49-58.















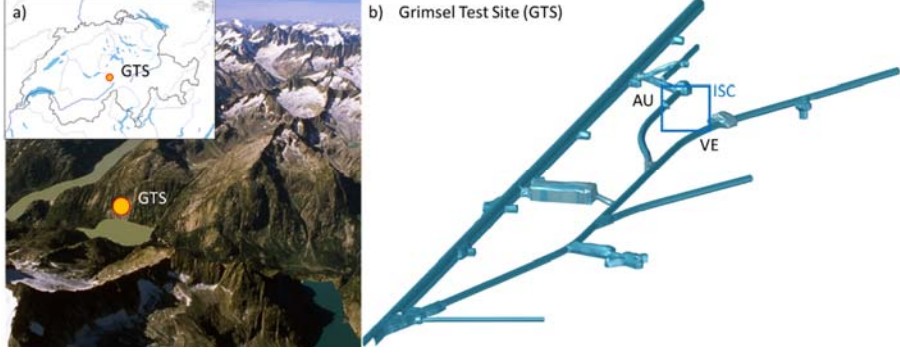


*Figure 1. a) Grimsel Test Site (GTS) is located in the Swiss Alps in the central part of Switzerland. b)*
*The in-situ stimulation and circulation experiment (ISC experiment) is implemented in the southern*
*part of the GTS in a low fracture density granitic rock*

**Pre-Stimulationsphase**

**Drilling**

**Stress measurements**

**Characterization**
- tunnel and core mapping
- geophysical borehole logs (OPTV, ATV, electrical resistivity, spectral gamma, full-wave sonic logs)
- hydraulic tests (i.e. single- and cross hole)
- geophysical characterization (i.e. GPR, active seismics, single- and cross hole and cross tunnel)
- tracer tests (dye, thermal tracer and DNA nanotracer)

**Monitoring**
- strain and tilt
- pore pressure
- temperature
- micro-seismics

**Stimulationsphase**

**Stimulation**
- stimulation of existing fractures and fault zone
- hydraulic fracturing in massive rock

**Monitoring**
- pressure und flow rates in active injection borehole
- pressure in passive injection borehole
- micro-seismicity in tunnels and boreholes
- pressure in boreholes and tunnel surface
- strain in boreholes and tunnel surface
- tilt at the tunnel surface
- dislocations in active injection borehole using an acoustic televiewer

**Post-Stimulationsphase**

**Characterization**
- geophysical borehole logs in the injection boreholes(electrical resistivity, spectral gamma, full-wave sonic logs)
- hydraulic tests (i.e. single- and cross hole)
- tracer tests (dye, thermal tracer und DNA nanotracer)
- geophysical characterization (i.e. GPR, active seismics, single- and cross hole and cross tunnel)


*Figure 2. The three test phases of the ISC experiments with listings of the main activities*
*during each phase.*

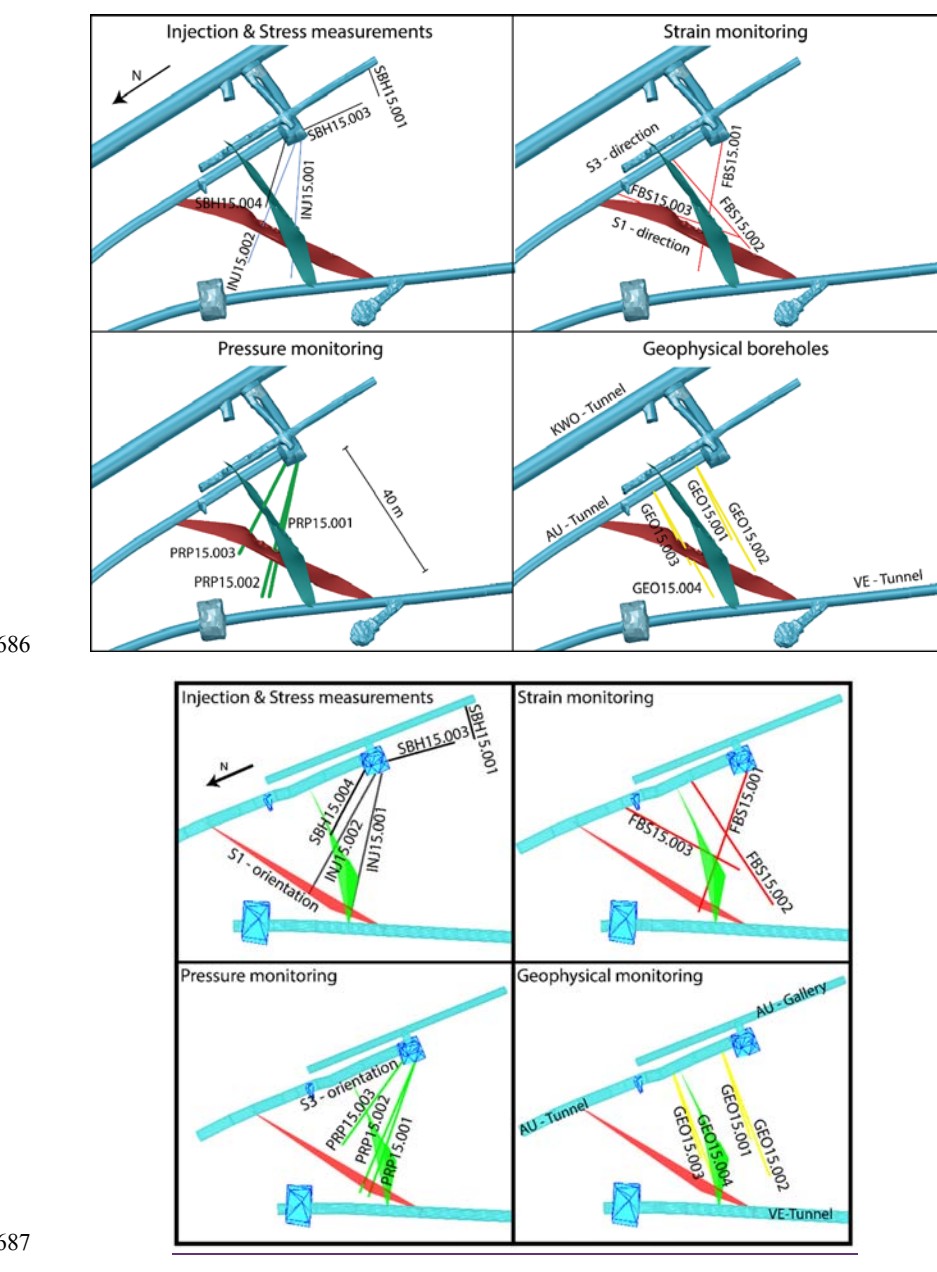



*Figure 3: The 15 boreholes drilled for the ISC experiment (view steeply inclined towards SE).*

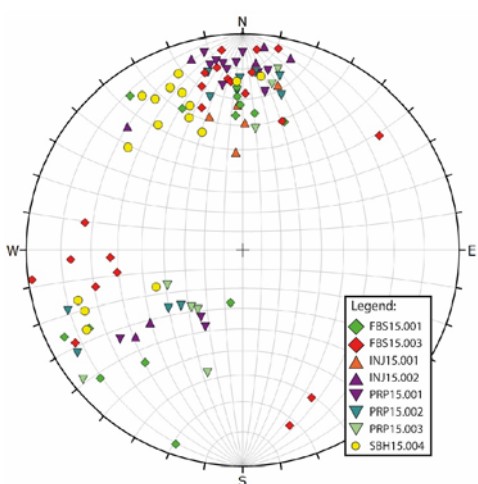


Figure 4. Brittle fractures between meta-basic dykes plotted into the lower hemisphere of a stereonet
plot. The data set is subdivided into the borehole where they were observed.







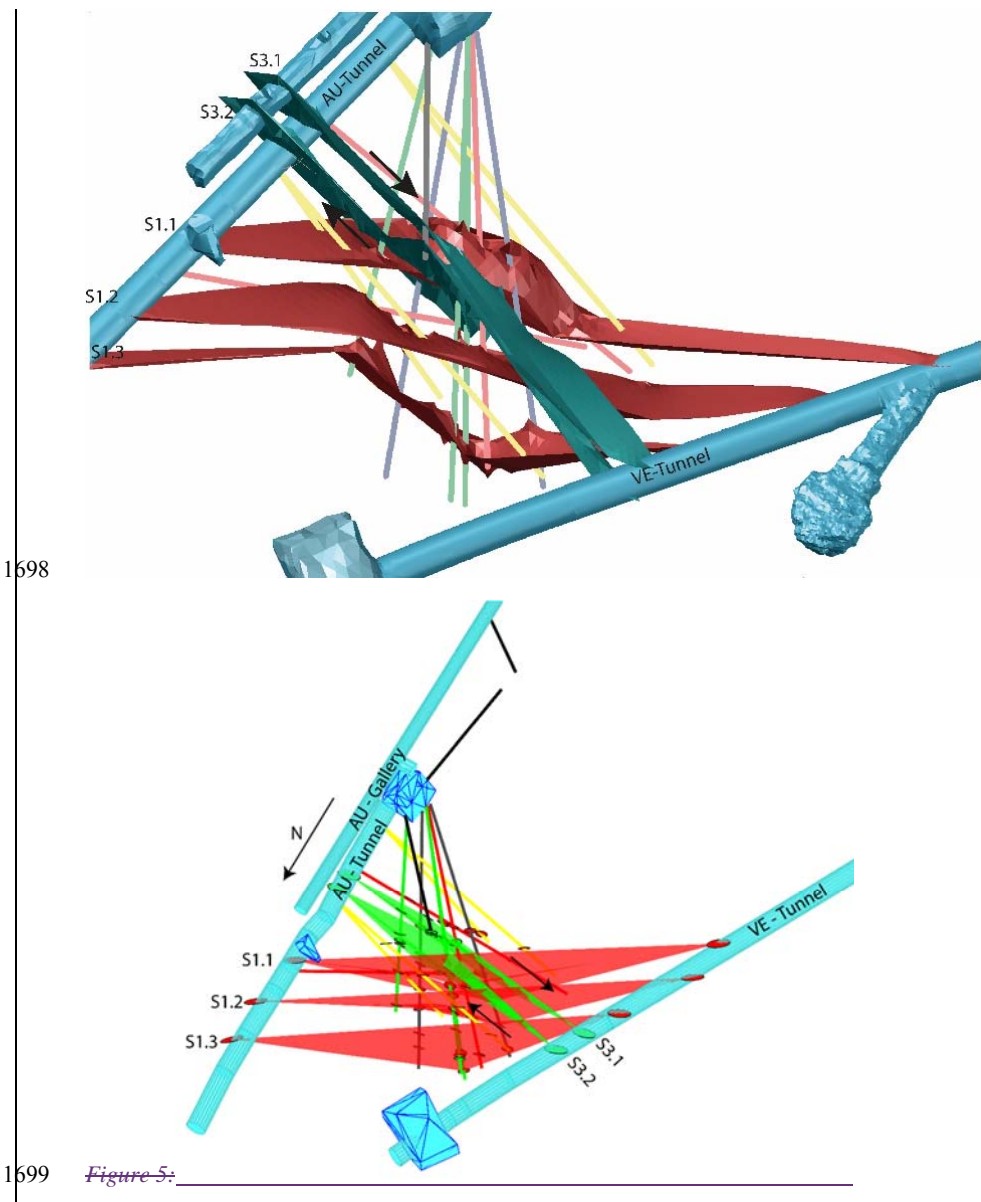



*Figure 4:* 3D-Model showing the boreholes drilled towards the rock volume for the in-situ stimulation
experiment, S1 (red) and S3 (~~blue~~green) oriented shear zones as well as the dextral shear sense at the
S3 shear zones indicated by the black arrows.



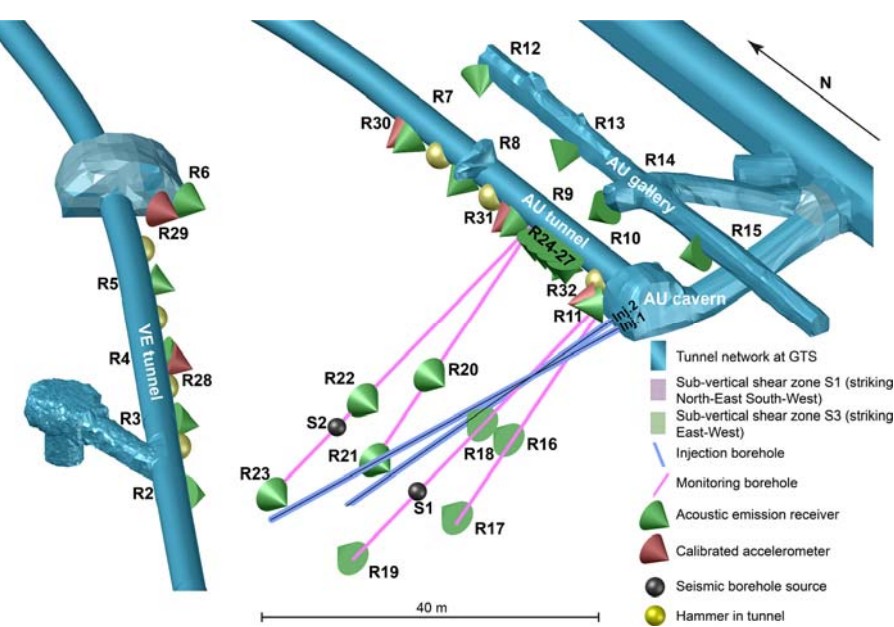

*Figure 65: Outline of seismic monitoring network including hammer sources and borehole*
*piezosources for active seismic surveys.*







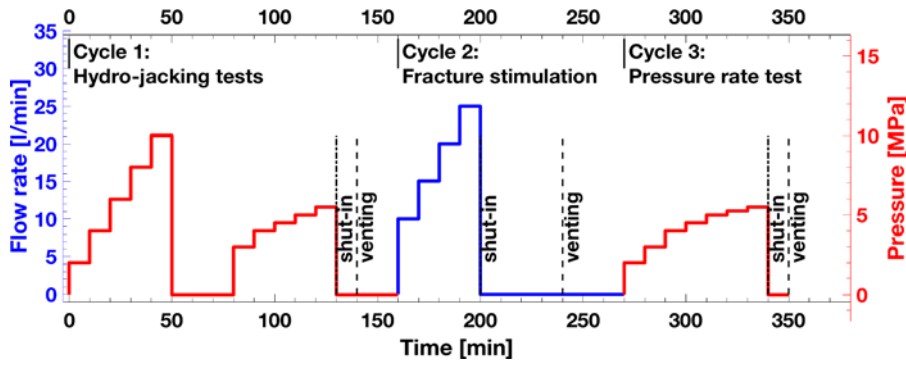

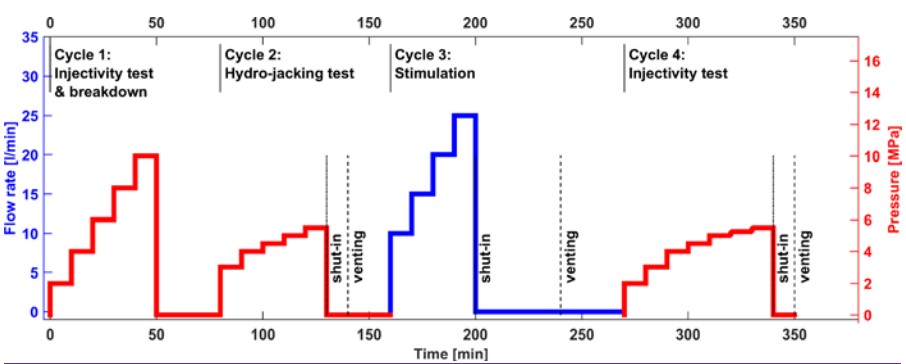

*Figure 7: Planned injection6: Injection protocol for hydroshearing experiments. Red curves denote pressure controlled injections. (Cycle 1), blue curves flow rate controlled injections (Cycle 2 and 3). The total volume injected is 1 m³.*

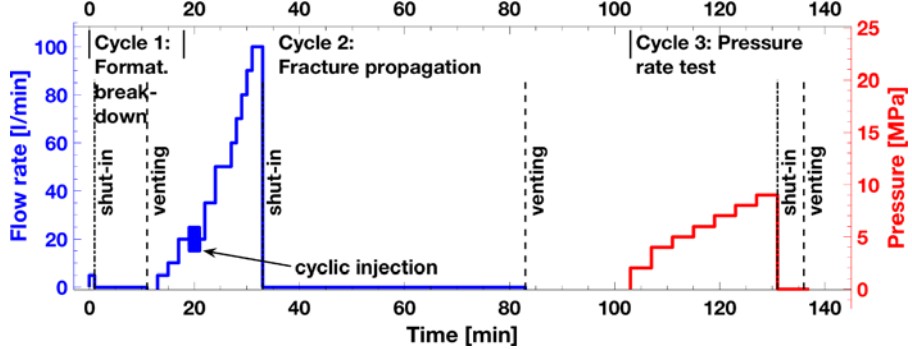

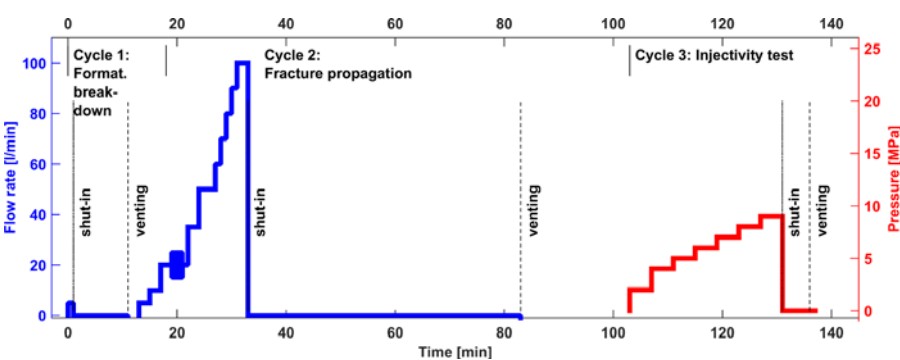

1723

Figure ~~8: Planned injection~~7: Injection protocol for hydrofracturing experiments. The blue
solid curve denotes flow rate controlled and the red solid curve pressure controlled injection.
~~The red dashed line respective the blue dashed line are the anticipated pressure respective flow~~
~~rate response.~~