# Peer review of "The seismo-hydro-mechanical behaviour during deep geothermal reservoir stimulations: open questions tackled in a decameter-scale in-situ stimulation experiment"

_Solid Earth, 2017_

## Referee Comment (RC1) · Anonymous Referee #1 · 18 Aug 2017

General comments

This is a review paper. It provides information on two aspects of hydraulic stimulation used for creating engineered geothermal systems (EGS):

I. extensive literature reviews on (i) the nature of the stimulation process and dedicated experiments on reservoir, intermediate and laboratory scale performed for enhancing low permeability of reservoir rocks; (ii) hydro-fracking experiments on reservoir, intermediate and laboratory scale performed for creating extensive fractures enabling flow

rates sufficient for extraction of relevant amounts of heat, and the associated rock mass deformation, seismic an aseismic slip, and induced seismicity.

II. A description of the scientific and experimental infrastructure in the Grimsel test site in the Swiss Alps implemented for the experiments to be performed in the In-situ Stimulation and Circulation Experiment

Part I is of great value for all present and future researchers in this field as it covers most if not all relevant work. Part II is probably intended to describe the infrastructure in a separate paper to be referenced by future papers describing and discussing the experiments currently under way and planned in the future. The value of combining these two aspects in one paper is not obvious. These are separate topics and would merit separate papers. Also, this would allow to go into more technical detail the second part. Here, it should be made more clear, which experiments are intended and which ones have been performed already. At first reading, it was a bit confusing discriminating between completed and planned experiments.

The m/s should be divided into two ones: I. the literature review. This part may stay more or less at it is; II: the description of the test site infrastructure: This would need to be revised an probably expanded for more technical detail.

---

## Referee Comment (RC2) · Anonymous Referee #2 · 8 Nov 2017

Review Comments

The seismo-hydro-mechanical behaviour during deep geothermal reservoir stimulations: open questions tackled in a decameter scale in-situ stimulation experiment, by Amann et al

General:

This manuscript has essentially two parts to it: The first is a review of coupled seismological-hydro-mechanical processes with a focus on experimental work across

scales. The second part is a fairly detailed description of a meso-scale controlled stimulation experiment conducted in a mine in the Swiss Alps. In some ways, this is an odd combination, and I am trying to convince myself that this combination is useful for the reader. This may be more a question for the editor of the journal rather than a decision to be made by a reviewer. Here are my main comments:

1) Part 1: Literature Review of seismo-hydro-mechanical processes I found the literature review well done and useful for a broad audience interested in EGS. Most of the relevant literature has been cited and adequately discussed. The manuscript is an interesting read for experts and non-experts in the field. I found this part of the paper acceptable for publication with some minor edits. The review culminates in a short section discussing open research questions and promoting meso-scale decameter experiments to bridge a scaled- and experimental gap existing between laboratory studies in large-scale field experiments.

2) Part 2: Description of the ISC experiment The second part of the manuscript is a description of the meso-scale decameter ISC experiment that the authors and partners conducted in 2017 in the Grimsel Rock Lab in the Swiss Alps. I am quite familiar with the experiment which was well designed and will down the road lead to many superb research advances. However, the description of ISC here falls short because it is largely a planning document describing the design and monitoring methods for the test, but not any results or findings. Plus, the description is at times painfully detailed (i.e., what manufacturer for sensors, which resolution, what frequency,….), leaving the reader to wonder why that these details are relevant given no results are shown. Also, the authors at times seem to be confused about the timing of writing the paper vs the timing of the test: while most text is in present tense ("The stimulation injections target natural fracture zones….", line 852) some phrases fall into past tense ("Selected stimulation intervals were isolated . . .", line 900). This is confusing, in particular since there is no early "warning" in Section 3 that the ISC description would not include any results or findings. Here is what I might suggest as remedy: - Make the second part much

shorter and higher level. I like that the literature review results in making a case for meso-scale experiments, which then is followed by the detailed description of the ISC. But for the purpose of his paper, the second part should be shorter and to the point. Also make sure that readers understand early on what to expect from the ISC section (i.e., no test results). - Or, my preferred option, provide some selected experimental results in Section 3, and defer for any further analyses in future papers. This would make the paper much more interesting. Selected results could be linked back to the research questions given on Page 20. (As you are adding experimental results, the design description parts of the ISC section should still be shortened considerably!)

Here are some other comments:

3) Lines 68-72 The description of EGS sounds like per definition these systems are only found at depth of at least 5 to 6 km, assuming a standard geothermal gradient like in Switzerland. But there are areas elsewhere in the world where the geothermal gradient is not standard, and where high temperatures are found shallower, and where EGS is needed because the necessary permeability is not there. In my opinion, EGS should be defined as a promising technology when there is sufficiently high geothermal temperature but not enough permeability to allow for heat harvesting. That can be a depth of 5-6 km or deeper, but can also be shallower.

4) Line 81-82 Since this is a review paper for the broad community, I would add a sentence or two explaining induced seismicity.

5) Line 105 Suggest adding "small (micro-seismic)" before induced seismic events are important monitoring….."

6) Line 124 "…experiment is currently being….". The term "currently" may be correct at the time of writing the manuscript, but this paper will be read in 10 years or longer into the future. Change to "has been conducted in 2017…"

7) Line 147 Change "envisaged" to "envisioned".

8) Line 147-148 Timeline may be off. The text discussed the early Hot Dry Rock work in the 70s and then notes that the design was adopted from oil and gas hydrofracture technology. Is this true? The current shale oil and gas hydro-fracturing with long horizontal wells was developed later, in the 90s and 2000s. But maybe I misunderstood?

9) Line 154 Add "new" before tensile fractures.

10) Line 208-209 Sentence structure. I suggest changing to: "The paucity of . . ..is largely a result of the considerable depth of typical geothermal resources (e.g., several kilometers), which. . ."

11) Line 207ff Going through the discussion of "Reservoir-scale experiments", I see various deep drilling and stimulation projects described, but no differentiation in terms of the rock types that were tested. Hydromechanical behavior is fundamentally different in crystalline vs argillite rocks, and this needs to be brought out here. Perhaps the same applies to other section of the manuscript; please check.

12) Line 231ff I suggest adding that when slip is mostly aseismic then the obvious yet important result is that micro-seismic measurements will not allow monitoring of slip.

13) Line 247 Add "deep" between 3.5 km and reservoir.

14) Section 2.1.2 What about the many relevant lab studies out of Penn State on hydroshearing (Marone, Elsworth).

15) Section 2.2 There is a dedicated section of "Hydraulic Fracturing". Should there also be a dedicated section on "Hydroshearing Experiments"? Or is that included in 2.1? If so, perhaps be more upfront about it because the title of Section 2.1 is "Stimulation Process".

16) Line 354. Instead of "remote" say: "at a distance of xxx from. . ."

17) Line 404-408. Please give specific reference for this sentence.

18) Line 465. Is it "van Ass" or "van As"?

19) Line 534-545. I suggest adding references to the work conducted for EGS sites in the U.S. at the Geysers.

20) Line 591. Change "question" to "questions"

21) Line 629 Same issue as before with the term "The ongoing. . .". The term "ongoing" may be correct at the time of writing the manuscript, but this paper will be read in 10 years or longer into the future.

22) Line 630 Consider changing "experiment tries to contribute to the filling. . .." with "experiment addresses this research gap". Same in Line 651: Change "is to contribute in finding . . .." To "is to find answers".

23) Section 3.2.1.1 As mentioned before, there is some going back and forth with present and past tense in the description of the experimental design, which is confusing. Consider present tense in Lines 708 and 709 for example (are drilled, are dedicated) vs. Lines 720 and 728 (was characterized, were used). Or look at lines 899-900 (the post stimulation phase) which is now back to past tense (were determined. . ., were isolated).

24) Section 3.2.1.2 ff Nice to see text here going back to research questions. Just numbering the research questions is a bit cumbersome for reader but I am not sure how to improve and verbalize the numbers.

25) Line 825 and 826 Very awkward use of present tense. Sentence "More detailed processing of the complete data is performed . . .." Is obviously wrong since a reference to the processing is given in the same sentence.

26) Lines 846ff How was the maximum injection volume determined? Simulations?

27) Line 852 Add "initial" before transmissivities.

28) Line 877-878 How did the team know that 5L/min would be sufficient to induce initial damage?

Please also note the supplement to this comment:
https://www.solid-earth-discuss.net/se-2017-79/se-2017-79-RC2-supplement.pdf
* * *

---

## Editor Comment (EC1) · C.M. Krawczyk (Editor) · 9 Nov 2017

Dear authors, your manuscript has been read and reviewed now by two reviewers and myself. It is a nicely readable and very suitable piece of work for SE. Since all reviewers came to the conclusion, that the combination of a review and a strong original experiment part in one manuscript is not convincing, please revise you paper accordingly. In this sense, I am happy to receive a fully elaborated review paper as revision, and I am confident that it will make its way in SE. With best regards, Lotte Krawczyk.

---

## Author Comment (AC1) · 1 Dec 2017

The seismo-hydro-mechanical behaviour during deep geothermal reservoir stimulations: open questions tackled in a decameter-scale in-situ stimulation experiment

Dear Editor, dear reviewers,

Thank you very much for the fast review process and the constructive and valuable comments on our manuscript. We are pleased to provide answers to the reviewer's comments and a substantially revised version of the above manuscript. The major

points of both reviewers were 1) the combination of a review and an original experiment part in one manuscript and 2) the timing of writing the second part of the paper versus the timing of the test. Both reviewers provided suggestions how to proceed with these two issues. Reviewer 1 suggests splitting the paper in a review paper and an experimental paper with much more details on the experimental design. Reviewer 2 suggests to substantially shortening the second part by providing much less details and higher-level information. The timing was straightforward to address, since meanwhile the experiment has been completed. The more severe issue during the revision process was associated with the question to split the paper into two separated papers. We decided not to split the paper in two parts, since a description of the experiment is important to discuss how we address the identified research questions. At the same time, the sensor and installation details will be published in subsequent papers. Thus, we decided to follow the advice of reviewer 2 and substantially shortened part two and report only about higher-level information of the experiment. We hope you find the revised manuscript suitable for publication in "Solid Earth".

Florian Amann

Reviewer 1:

This is a review paper. It provides information on two aspects of hydraulic stimulation used for creating engineered geothermal systems (EGS):

I. extensive literature reviews on (i) the nature of the stimulation process and dedicated experiments on reservoir, intermediate and laboratory scale performed for enhancing low permeability of reservoir rocks; (ii) hydro-fracking experiments on reservoir, intermediate and laboratory scale performed for creating extensive fractures enabling flowrates sufficient for extraction of relevant amounts of heat, and the associated rock mass deformation, seismic an aseismic slip, and induced seismicity.

II. A description of the scientific and experimental infrastructure in the Grimsel test site in the Swiss Alps implemented for the experiments to be performed in the In-situ

[Figure]

Stimulation and Circulation Experiment Part I is of great value for all present and future researchers in this field as it covers most if not all relevant work. Part II is probably intended to describe the infrastructure in a separate paper to be referenced by future papers describing and discussing the experiments currently under way and planned in the future. The value of combining these two aspects in one paper is not obvious. These are separate topics and would merit separate papers. Also, this would allow to go into more technical detail the second part. Here, it should be made more clear, which experiments are intended and which ones have been performed already. At first reading, it was a bit confusing discriminating between completed and planned experiments.

The m/s should be divided into two ones: I. the literature review. This part may stay more or less at it is; II: the description of the test site infrastructure: This would need to be revised a probably expanded for more technical detail.

Answer: We agree with the reviewer that the second part of our paper did not consort very well with the review part. We also agree that the timing of writing was confusing. We thus shortened the second part substantially and made clear that all parts of the experiments are completed. We prefer not to split the paper into two separate papers, with more details on the experimental infrastructure. This is because the results of the experiments will be published with detailed descriptions of the various infrastructure components.

---

## Author Comment (AC2) · 1 Dec 2017

The seismo-hydro-mechanical behaviour during deep geothermal reservoir stimulations: open questions tackled in a decameter-scale in-situ stimulation experiment

Dear Editor, dear reviewers,

Thank you very much for the fast review process and the constructive and valuable comments on our manuscript. We are pleased to provide answers to the reviewer's comments and a substantially revised version of the above manuscript. The major

points of both reviewers were 1) the combination of a review and an original experiment part in one manuscript and 2) the timing of writing the second part of the paper versus the timing of the test. Both reviewers provided suggestions how to proceed with these two issues. Reviewer 1 suggests splitting the paper in a review paper and an experimental paper with much more details on the experimental design. Reviewer 2 suggests to substantially shortening the second part by providing much less details and higher-level information. The timing was straightforward to address, since meanwhile the experiment has been completed. The more severe issue during the revision process was associated with the question to split the paper into two separated papers. We decided not to split the paper in two parts, since a description of the experiment is important to discuss how we address the identified research questions. At the same time, the sensor and installation details will be published in subsequent papers. Thus, we decided to follow the advice of reviewer 2 and substantially shortened part two and report only about higher-level information of the experiment. We hope you find the revised manuscript suitable for publication in "Solid Earth".

Florian Amann

Reviewer 2:

General:

This manuscript has essentially two parts to it: The first is a review of coupled seismological-hydro-mechanical processes with a focus on experimental work across scales. The second part is a fairly detailed description of a meso-scale controlled stimulation experiment conducted in a mine in the Swiss Alps. In some ways, this is an odd combination, and I am trying to convince myself that this combination is useful for the reader. This may be more a question for the editor of the journal rather than a decision to be made by a reviewer. Here are my main comments:

1) Part 1: Literature Review of seismo-hydro-mechanical processes

I found the literature review well done and useful for a broad audience interested in EGS. Most of the relevant literature has been cited and adequately discussed. The manuscript is an interesting read for experts and non-experts in the field. I found this part of the paper acceptable for publication with some minor edits. The review culminates in a short section discussing open research questions and promoting meso-scale decameter experiments to bridge a scaled- and experimental gap existing between laboratory studies in large-scale field experiments.

2) Part 2: Description of the ISC experiment

The second part of the manuscript is a description of the meso-scale decameter ISC experiment that the authors and partners conducted in 2017 in the Grimsel Rock Lab in the Swiss Alps. I am quite familiar with the experiment which was well designed and will down the road lead to many superb research advances. However, the description of ISC here falls short because it is largely a planning document describing the design and monitoring methods for the test, but not any results or findings. Plus, the description is at times painfully detailed (i.e., what manufacturer for sensors, which resolution, what frequency,….), leaving the reader to wonder why that these details are relevant given no results are shown. Also, the authors at times seem to be confused about the timing of writing the paper vs the timing of the test: while most text is in present tense ("The stimulation injections target natural fracture zones….", line 852) some phrases fall into past tense ("Selected stimulation intervals were isolated . . .", line 900). This is confusing, in particular since there is no early "warning" in Section 3 that the ISC description would not include any results or findings. Here is what I might suggest as remedy:

- Make the second part much shorter and higher level. I like that the literature review results in making a case for meso-scale experiments, which then is followed by the detailed description of the ISC. But for the purpose of his paper, the second part should be shorter and to the point. Also make sure that readers understand early on what to expect from the ISC section (i.e., no test results).

- Or, my preferred option, provide some selected experimental results in Section 3, and defer for any further analyses in future papers. This would make the paper much more interesting. Selected results could be linked back to the research questions given on Page 20. (As you are adding experimental results, the design description parts of the ISC section should still be shortened considerably!)

We shortened the second part substantially and made clear that all parts of the experiments are completed. We also included an "early warning" for the reader that results are not discussed in this paper. Through shortening the second part of the paper we omitted unnecessary detail and focused on higher-level information.

Here are some other comments:

3) Lines 68-72

The description of EGS sounds like per definition these systems are only found at depth of at least 5 to 6 km, assuming a standard geothermal gradient like in Switzerland. But there are areas elsewhere in the world where the geothermal gradient is not standard, and where high temperatures are found shallower, and where EGS is needed because the necessary permeability is not there. In my opinion, EGS should be defined as a promising technology when there is sufficiently high geothermal temperature but not enough permeability to allow for heat harvesting. That can be a depth of 5-6 km or deeper, but can also be shallower.

We reworded this part so that it is clearer that EGS are mostly applied to regions with not enough permeability.

4) Line 81-82

Since this is a review paper for the broad community, I would add a sentence or two explaining induced seismicity.

We added some more explanation why induced seismicity occurs and why it is problematic.

5) Line 105

Suggest adding "small (micro-seismic)" before induced seismic events are important monitoring….."

Added.

6) Line 124

"…experiment is currently being….". The term "currently" may be correct at the time of writing the manuscript, but this paper will be read in 10 years or longer into the future. Change to "has been conducted in 2017…"

Corrected.

7) Line 147

Change "envisaged" to "envisioned".

Changed.

8) Line 147-148

Timeline may be off. The text discussed the early Hot Dry Rock work in the 70s and then notes that the design was adopted from oil and gas hydrofracture technology. Is this true? The current shale oil and gas hydro-fracturing with long horizontal wells was developed later, in the 90s and 2000s. But maybe I misunderstood?

Hydraulic fracturing was already used in the 1940s in the conventional oil and gas industry. Montgomery and Smith (2010) showed that hydraulic fracturing was used first time in 1947 in the Hugoton gas field in Kansas to enhance the production.

C. T. Montgomery, M. B. Smith: Hydraulic Fracturing. History of an enduring technology. In: JPT 62/12. 2010

9) Line 154

Add "new" before tensile fractures.

Changed.

10) Line 208-209

Sentence structure. I suggest changing to: "The paucity of . . ..is largely a result of the considerable depth of typical geothermal resources (e.g., several kilometers), which. . ."

Changed.

11) Line 207ff

Going through the discussion of "Reservoir-scale experiments", I see various deep drilling and stimulation projects described, but no differentiation in terms of the rock types that were tested. Hydromechanical behavior is fundamentally different in crystalline vs argillite rocks, and this needs to be brought out here. Perhaps the same applies to other section of the manuscript; please check.

We added a statement indicating that the seismo-hydro-mechanical response also depends on rock type and we also indicate rock type where a case study is explicitly described.

12) Line 231ff

I suggest adding that when slip is mostly aseismic then the obvious yet important result is that micro-seismic measurements will not allow monitoring of slip.

We added this.

13) Line 247

Add "deep" between 3.5 km and reservoir.

Added.

14) Section 2.1.2

What about the many relevant lab studies out of Penn State on hydroshearing (Marone, Elsworth).

Indeed there are many interesting studies coming from this lab. Although we kept the laboratory part short on purpose (as it would be worth a literature review on its own), we added some links to these experiments where they contribute to the concept of the section.

15) Section 2.2

There is a dedicated section of "Hydraulic Fracturing". Should there also be a dedicated section on "Hydroshearing Experiments"? Or is that included in 2.1? If so, perhaps be more upfront about it because the title of Section 2.1 is "Stimulation Process".

We changed the titles to '2.1 Stimulation by hydraulic shearing' and '2.2 Stimulation by hydraulic fracturing'.

16) Line 354.

Instead of "remote" say: "at a distance of xxx from..."

Changed.

17) Line 404-408.

Please give specific reference for this sentence.

We added reference to Jeffrey et al., (2009).

18) Line 465.

Is it "van Ass" or "van As"?

It is 'van As'. Changed.

19) Line 534-545.

I suggest adding references to the work conducted for EGS sites in the U.S. at the

Geysers.

We added some of the recent work at the Geysers EGS demonstration project.

20) Line 591.

Change "question" to "questions"

Changed.

21) Line 629

Same issue as before with the term "The ongoing...". The term "ongoing" may be correct at the time of writing the manuscript, but this paper will be read in 10 years or longer into the future.

Changed.

22) Line 630

Consider changing "experiment tries to contribute to the filling...." with "experiment addresses this research gap". Same in Line 651: Change "is to contribute in finding ...." To "is to find answers".

Reworded as suggested.

23) Section 3.2.1.1

As mentioned before, there is some going back and forth with present and past tense in the description of the experimental design, which is confusing. Consider present tense in Lines 708 and 709 for example (are drilled, are dedicated) vs. Lines 720 and 728 (was characterized, were used). Or look at lines 899-900 (the post stimulation phase) which is now back to past tense (were determined..., were isolated).

We agree and we made major changes

24) Section 3.2.1.2 ff

Nice to see text here going back to research questions. Just numbering the research questions is a bit cumbersome for reader but I am not sure how to improve and verbalize the numbers.

We agree. In addition to indicating the link to a research question by referring to a number, we also added in short words what the corresponding research question is.

25) Line 825 and 826

Very awkward use of present tense. Sentence "More detailed processing of the complete data is performed . . .." Is obviously wrong since a reference to the processing is given in the same sentence.

We changes the entire section and omitted this section

26) Lines 846ff

How was the maximum injection volume determined? Simulations?

The maximum volume was determined in a detailed risk analysis prior to the experiment. We added more explanation and added a reference to the risk study. In the risk study, we used the scaling law between injection volume and maximum earthquake magnitude as well as the maximum allowable ground acceleration for the tunnel and infrastructure.

27) Line 852

Add "initial" before transmissivities.

done

28) Line 877-878

How did the team know that 5L/min would be sufficient to induce initial damage.

The rock mass is very tight and little flow rates will cause an immediate pressure build-up that cause hydraulic fracturing initiation. We performed hydraulic fracturing

experiments for stress determination prior to the stimulation phases and gained some experience through these tests.
* * *